# RISK-SENSITIVE RL FOR ALLEVIATING EXPLORATION DILEMMAS IN LARGE LANGUAGE MODELS

**Yuhua Jiang**[1]* **Jiawei Huang**[2]* **Yufeng Yuan**[3] **Xin Mao**[3]
**Yu Yue**[3] **Qianchuan Zhao**[1] **Lin Yan**[3]
[1]Tsinghua University  [2]ETH Zurich  [3]ByteDance Seed

## ABSTRACT

Reinforcement Learning with Verifiable Rewards (RLVR) has proven effective for enhancing Large Language Models (LLMs) on complex reasoning tasks. However, existing methods suffer from an *exploration dilemma*: the sharply peaked initial policies of pre-trained LLMs confine standard RL algorithms to a narrow set of solutions, boosting single-solution accuracy (pass@1) but suppressing solution diversity and multi-solution performance (pass@k). As a result, RLVR often distills existing capabilities rather than discovering new reasoning strategies. To overcome this, we introduce a *Risk-Sensitive Reinforcement Learning* framework. Our approach employs a risk-seeking objective that interpolates between mean and maximum rewards, leading to a novel algorithm, Risk-Sensitive GRPO (RS-GRPO), which drives deeper exploration by amplifying learning from challenging prompts. Remarkably, RS-GRPO is simple to implement, requiring only minor code modifications. On six mathematical reasoning benchmarks and with five different LLMs, RS-GRPO consistently improves pass@k performance while maintaining or enhancing pass@1 accuracy.

$$\hat{\nabla}_\theta \mathcal{J}(\pi_\theta) = \frac{1}{N}\sum_{i=1}^{N}\hat{A}^{\pi_\theta}(y_i)\nabla_\theta \log \pi_\theta(y_i \mid x), \quad \hat{A}^{\pi_\theta}(y_i) = r(y_i) - \frac{1}{N}\sum_{i=1}^{N} r(y_j)$$

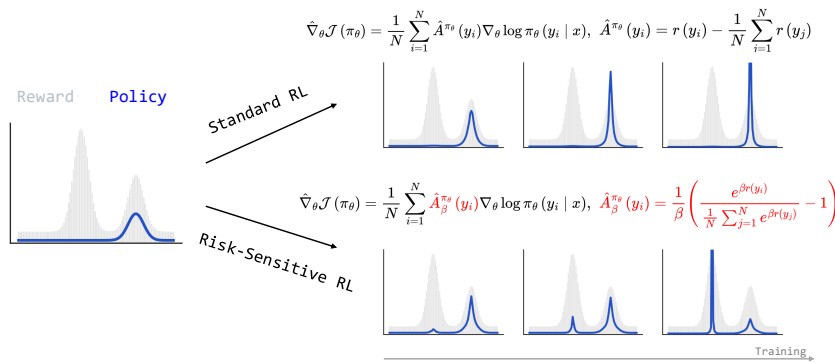

$$\hat{\nabla}_\theta \mathcal{J}(\pi_\theta) = \frac{1}{N}\sum_{i=1}^{N}\hat{A}_\beta^{\pi_\theta}(y_i)\nabla_\theta \log \pi_\theta(y_i \mid x), \quad \hat{A}_\beta^{\pi_\theta}(y_i) = \frac{1}{\beta}\left(\frac{e^{\beta r(y_i)}}{\frac{1}{N}\sum_{j=1}^{N} e^{\beta r(y_j)}} - 1\right)$$

Figure 1: Starting from a sharply peaked policy whose probability mass is concentrated on suboptimal rewards, standard RL further collapses the policy and fails to converge to the optimum, whereas risk-sensitive RL promotes exploration and converges to the optimal solution. The training configuration is described in Section 3.1.

## 1 INTRODUCTION

Reinforcement learning (RL) with verifiable rewards has recently emerged as a highly effective paradigm for enhancing large language models (LLMs) in complex reasoning domains, enabling models to achieve superhuman performance (Jaech et al., 2024; Guo et al., 2025; Kimi et al., 2025; Comanici et al., 2025; Seed et al., 2025). However, a growing body of evidence reveals a critical failure mode in this approach, which we term the **exploration dilemma**: while current RL methods improve average accuracy (pass@1), they often achieve this by simply sharpening the policy distribution around a limited number of homogeneous solutions. This concentration of probability mass leads to a collapse in solution diversity, causing performance on the more general pass@k

---

*Work done at ByteDance Seed. Corresponding to `jiangyh22@mails.tsinghua.edu.cn`

metric to stagnate or even degrade compared to the base model (Yue et al., 2025; Wu et al., 2025; He et al., 2025a; Liu et al., 2025a; Shah et al., 2025). Rather than discovering genuinely novel reasoning strategies, existing methods merely reinforce pre-trained biases and fail to expand the policy's capability frontier, posing a significant bottleneck to progress.

We argue this dilemma arises from a fundamental mismatch between the optimization landscape of LLMs and the dynamics of standard RL algorithms. In contrast to traditional RL settings (e.g., game playing (Mnih et al., 2015; Silver et al., 2017)) where training starts from a randomly initialized policy, LLMs begin with a highly specialized policy distribution that is already sharply peaked around certain solutions. If those initial peaks are not supported in the regions that yield optimal rewards, standard RL optimizers face a significant challenge: they struggle to escape the gravitational pull of the pretrained model's biases and tend to converge to a nearby, but often suboptimal mode (Wu et al., 2025; He et al., 2025a). This prevents the discovery of more diverse and powerful reasoning paths.

To address this limitation, we introduce a **Risk-Sensitive RL** framework designed to enhance exploration in LLM training, enabling policies to escape local optima induced by the initial bias. Our core idea is to replace the standard risk-neutral objective, which optimizes the mean of the reward distribution, with a risk-seeking one that instead interpolates smoothly between the mean and the maximum reward. By employing an exponential utility function, we derive a new formulation of policy gradient with a corresponding risk-sensitive advantage. This advantage function dynamically re-weights optimization, placing greater emphasis on hard prompts where the model performs poorly, thereby driving the policy to explore under-explored regions of the solution space.

Our approach is instantiated as a simple yet powerful algorithm Risk-Sensitive GRPO (RS-GRPO), which can be implemented as a drop-in replacement for the advantage calculation in standard RL for LLM pipelines. Through extensive experiments on six mathematical reasoning benchmarks with a diverse set of six LLMs, we demonstrate that RS-GRPO consistently and significantly improves pass@k performance over both the base models and the standard GRPO baseline. Crucially, RS-GRPO achieves this while maintaining or even improving pass@1 accuracy, striking a more effective balance than prior methods. **Our main contributions are summarized as follows**:

- We introduce a risk-sensitive RL framework to address the exploration dilemma in LLM fine-tuning and instantiate it as a simple yet powerful algorithm, Risk-Sensitive GRPO (RS-GRPO). (Section 2.)

- We provide theoretical and empirical evidence that standard RL often fails to reach the global optimum when the initial policy is sharply peaked and far from optimal, whereas our risk-sensitive formulation avoids this pitfall. (Section 3.)

- We demonstrate through large-scale experiments on mathematical reasoning that RS-GRPO significantly improves pass@k performance while maintaining or even enhancing pass@1 accuracy, achieving a superior trade-off compared to existing methods. (Section 4.)

## 1.1 RELATED WORK

**RL Exploration** Exploration remains a central challenge in reinforcement learning (RL), but its nature differs significantly between traditional applications and LLMs. In domains like game-playing, where policies are often trained from random initializations, broad exploration is essential and often encouraged by intrinsic motivation based on state novelty (Oudeyer et al., 2007; Bellemare et al., 2016; Pathak et al., 2017; Burda et al., 2018; Henaff et al., 2022; Yang et al., 2024b; Jiang et al., 2025a). While some have adapted intrinsic motivation to LLMs (Bai et al., 2025; Gao et al., 2025), they often introduce auxiliary networks, complicating training and scaling.

The most direct method to encourage exploration in LLMs is to maximize policy entropy as an auxiliary objective, but its effectiveness can be limited (Cui et al., 2025; Chen et al., 2025), spurring research into alternative directions. Some approaches focus on enhancing the reasoning process through self-reflection (Jiang et al., 2025b; Kumar et al., 2024; Ma et al., 2025a; Yeo et al., 2025), while others investigate policy entropy dynamics to prevent mode collapse (Yu et al., 2025; Cui et al., 2025; Cheng et al., 2025). Orthogonal to these methods, our work contributes to a line of research focused on directly optimizing for inference-time objectives. This can be viewed as a form of risk-sensitive learning, where early efforts on Best-of-N (BoN) alignment (Gui et al., 2024; Amini et al., 2025; Chow et al., 2025; Balashankar et al., 2025) have evolved into policy gradient methods for pass@k optimization. Notable developments include various policy gradient formulations for pass@k (Tang et al., 2025; Walder & Karkhanis, 2025; Mahdavi et al., 2025; Chen et al., 2025). As

Table 1: **Comparison of Pass@k optimization methods with Risk-Sensitive RL.**

| Methods | Binary Rewards | Continuous Rewards | Dense Signal |
|---|---|---|---|
| Tang et al. (2025) | ✓ | ✓ | ✗ |
| Walder & Karkhanis (2025) | ✓ | ✓ | ✗ |
| Mahdavi et al. (2025) | ✓ | ✗ | ✗ |
| Chen et al. (2025) | ✓ | ✗ | ✗ |
| Risk-Sensitive (ours) | ✓ | ✓ | ✓ |

shown in Table 1 and detailed in Appendix D, our risk-sensitive framework offers two key advantages over these pass@k optimization methods. First, our formulation naturally handles continuous rewards, whereas prior methods are often restricted to binary signals (Chen et al., 2025; Mahdavi et al., 2025). Second, our method yields a denser advantage signal. In most of pass@k methods, the optimization weight vanishes once prompt accuracy surpasses a given threshold (e.g., $1 - \frac{k}{N}$), which can hinder Pass@1 improvement. Our risk-sensitive formulation, by contrast, sustains a non-zero gradient even for high-accuracy prompts, thereby facilitating a more effective trade-off between Pass@k and Pass@1 performance. Detailed comparsion can found in Appendix D.

**Risk-Sensitive RL** Risk-sensitive RL (Howard & Matheson, 1972; Heger, 1994; Neuneier & Mihatsch, 1998; Mihatsch & Neuneier, 2002) aims to model and manage the risks associated with decision-making under uncertainty, moving beyond the standard expectation-based objective. While early work focused on risk-averse strategies for safety-critical domains such as financial trading (Filos, 2019), energy storage (Liu et al., 2024), and robotics (Nass et al., 2019; Noorani et al., 2022; Shi et al., 2024), the advent of distributional RL (Bellemare et al., 2017) has enabled more nuanced approaches. This paradigm facilitates not only risk-averse but also risk-seeking behaviors, which have been shown to promote exploration in domains like game playing (Jiang et al., 2024; Ma et al., 2025b; Mavrin et al., 2019). Our work posits that risk-seeking optimization is critical for escaping the sharply peaked initial policy distribution of pre-trained models and enabling broader exploration.

## 1.2 BACKGROUND

We formulate language generation as a reinforcement learning (RL) problem. A language model acts as a policy $\pi_\theta$, which generates a response $y$ to a prompt $x$ with probability $\pi_\theta(y|x)$. The quality of each response is measured by a reward function $r(x, y)$. The standard objective is to maximize expected reward:

$$\mathcal{J}(\pi_\theta) = \mathbb{E}_{x \sim \mathcal{D}, y \sim \pi_\theta(\cdot|x)}[r(x, y)]. \tag{1}$$

This objective is typically optimized via policy gradient, as stated by:

$$\nabla_\theta \mathcal{J}(\pi_\theta) = \mathbb{E}_{x \sim \mathcal{D}, y \sim \pi_\theta(\cdot|x)}[A^{\pi_\theta}(y)\nabla_\theta \log \pi_\theta(y|x)], \tag{2}$$

where $A^{\pi_\theta}$ denotes the advantage function. Empirically, for each prompt $x$, we sample $N$ responses $\{y_i\}_{i=1}^N$ from $\pi_\theta(\cdot|x)$ and construct stochastic gradient estimates:

$$\hat{\nabla}_\theta \mathcal{J}(\pi_\theta) = \frac{1}{N} \sum_{i=1}^N \hat{A}^{\pi_\theta}(y_i)\nabla_\theta \log \pi_\theta(y_i|x), \tag{3}$$

where $\hat{A}^{\pi_\theta}(y_i) = r(y_i) - \frac{1}{N}\sum_{j=1}^N r(y_j)$, with $y_j \sim \pi_\theta(\cdot|x)$, is the advantage estimate [1] For clarity, we omit terms common in RLHF, such as regularization and importance sampling, and drop the explicit dependency on $x$ when unambiguous.

**Pass@k.** Pass@k (Chen et al., 2021; Kulal et al., 2019) estimates the probability that at least one of $k$ generated responses is correct. It serves as a key inference-time objective, reflecting exploration ability and approximating best-of-$k$ under a reliable reward model:

$$\text{Pass@k} = \mathbb{E}_{\{y_i\}_{i=1}^k \sim \pi_\theta}\left[\max\left(r(y_1), \ldots, r(y_k)\right)\right]. \tag{4}$$

---

[1]We omit the standard deviation normalization used in the original GRPO (Shao et al., 2024) algorithm, which, as noted by DrGRPO (Liu et al., 2025b), introduces bias.

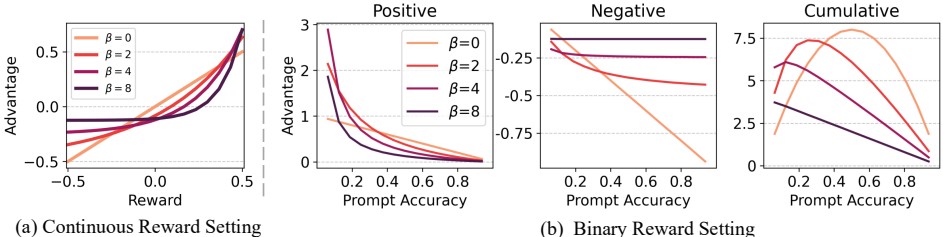

Figure 2: Analysis of the risk-sensitive advantage function with varying risk-sensitivity $\beta$.

## 2 RISK-SENSITIVE REINFORCEMENT LEARNING

**Desiderata.** Standard average reward optimization is often insufficient for tasks where exploration is critical, particularly when the initial policy distribution is sharply peaked. To this end, we design a new objective to promote exploration, for which we establish two primary desiderata. First, it should value all high-reward outcomes, not just the most probable one, thereby moving beyond simple mean-reward maximization and toward a maximum-reward-seeking objective. Second, the objective should provide a flexible and principled mechanism to interpolate between optimizing for the mean reward and the maximum reward, balancing exploration and exploitation. We find that *risk-sensitive RL* provides a natural framework for designing such an objective.

**Objective.** To meet these desiderata, we employ the risk-sensitive objective derived from exponential utility (Howard & Matheson, 1972). This objective provides a principled way to control the trade-off between exploration and exploitation. For a given policy $\pi_\theta$ and prompt $x$, the risk-sensitive objective is defined as:

$$\mathcal{J}_{\text{RS}}(\pi_\theta) = \mathbb{E}_{x \sim \mathcal{D}}\left[\frac{1}{\beta}\log\mathbb{E}_{y \sim \pi_\theta(\cdot|x)}\left[e^{\beta r(y)}\right]\right], \tag{5}$$

where the hyperparameter $\beta \in \mathbb{R}$ controls the risk-sensitivity level:

- *Risk-Neutral ($\beta \to 0$):* Recovers the standard expected reward, $\mathbb{E}[r(y)]$.
- *Risk-Seeking ($\beta \to +\infty$):* Approaches the maximum reward, $\max_y r(y)$, encouraging exploration.
- *Risk-Averse ($\beta \to -\infty$):* Approaches the minimum reward, $\min_y r(y)$, promoting robustness.

To effectively explore the solution space, we adopt a risk-seeking objective ($\beta > 0$). As $\beta$ increases, the objective places greater weight on high-reward outcomes, smoothly interpolating from the mean to the maximum reward. As $\beta \to 0$, it recovers the standard mean-reward objective. We now derive the corresponding policy gradient.

### 2.1 POLICY GRADIENT FOR THE RISK-SENSITIVE OBJECTIVE

We first derive the risk-sensitive policy gradient in theorem below, and defer its proof to Appx. C.

**Theorem 1.** *The policy gradient of the risk-sensitive objective in Eq.* (5) *is given by*

$$\nabla_\theta \mathcal{J}_{RS}(\pi_\theta) = \mathbb{E}_{x \sim \mathcal{D},\, y \sim \pi_\theta(\cdot|x)}\left[A_\beta^{\pi_\theta}(y)\nabla_\theta \log \pi_\theta(y \mid x)\right], \tag{6}$$

*where the risk-sensitive advantage $A_\beta^{\pi_\theta}$ is*

$$A_\beta^{\pi_\theta}(y) = \frac{1}{\beta}\left(\frac{e^{\beta r(y)}}{\mathbb{E}_{y' \sim \pi_\theta(\cdot|x)}[e^{\beta r(y')}]} - 1\right). \tag{7}$$

**Practical Implementation** In practice, we approximate the gradient for each prompt $x$ using $N$ samples $\{y_i\}_{i=1}^N \sim \pi_\theta(\cdot|x)$, yielding $\hat{\nabla}_\theta \mathcal{J}_x(\pi_\theta) = \frac{1}{N}\sum_{i=1}^N \hat{A}_\beta^{\pi_\theta}(y_i)\nabla_\theta \log \pi_\theta(y_i \mid x)$, where the empirical advantage is defined as

$$\hat{A}_\beta^{\pi_\theta}(y_i) = \frac{1}{\beta}\left(\frac{e^{\beta r(y_i)}}{\frac{1}{N}\sum_{j=1}^N e^{\beta r(y_j)}} - 1\right). \tag{8}$$

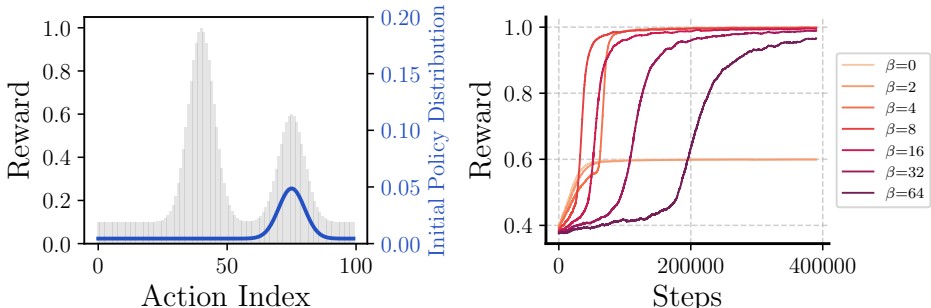

Figure 3: **A bandit experiment demonstrating that risk-sensitive RL can escape a local optimum that traps its standard RL counterpart. Left:** The reward landscape shows a global optimum and a distinct local optimum where the policy is initialized. **Right:** A standard risk-neutral policy ($\beta = 0$) is trapped locally, while risk-sensitive policies ($\beta \geq 4$) converge to the global optimum.

A key feature of this formulation is that it only alters the advantage computation while leaving the policy gradient structure intact. This allows our risk-sensitive advantage to serve as a drop-in replacement in existing GRPO-based RL algorithms (Shao et al., 2024; Yu et al., 2025; Liu et al., 2025b), requiring only minimal code modifications.

## 2.2 ANALYSIS OF THE RISK-SENSITIVE ADVANTAGE FUNCTION

The effectiveness of our method stems from the behavior of the risk-sensitive advantage function. To analyze its properties, we examine how the distribution of advantage values changes with the risk-sensitivity level, $\beta$, as illustrated in Figure 2.

**Continuous Reward Setting.** As shown in Figure 2(a), in a continuous reward space, the standard policy gradient ($\beta \to 0$) yields an advantage that is linear with the reward. As $\beta$ increases, the function sharpens into a step-like curve. This transformation amplifies the advantage for high-reward samples and suppresses it for low-reward ones.

**Binary Reward Setting.** Figure 2(b) illustrates the advantage dynamics in a binary reward setting (common in RLVR) as a function of prompt accuracy—the fraction of correct responses out of 16 samples. As $\beta$ increases, the advantage function increasingly prioritizes correct responses on hard prompts (low accuracy) while reducing the penalty for incorrect ones on easy prompts (high accuracy), as seen in the *Positive* and *Negative* subplots. Consequently, the *Cumulative* plot shows that the total advantage magnitude per prompt (sum of absolute advantages) shifts from peaking at 50%-accuracy prompts (for $\beta \to 0$) toward lower-accuracy ones. This demonstrates that as $\beta$ increases, risk-sensitive RL re-weights the advantage signals to prioritize harder prompts.

## 3 WHY RISK-SENSITIVE RL IS BETTER

Fine-tuning LLMs with RL often starts from a sharply peaked pretrained policy. Standard mean-reward optimization methods can be trapped in local optima corresponding to high-probability regions of this initial distribution, failing to discover global optima in low-probability areas. The risk-sensitive approach we adopt is designed to overcome this limitation. This section provides both empirical and theoretical support for this claim.

### 3.1 EMPIRICAL PERSPECTIVE

To illustrate the exploration dilemma, we design a 100-armed bandit problem where each action yields a deterministic reward. Figure 3a visualizes the reward landscape, which features two distinct peaks: a global optimum (reward 1.0) and a prominent suboptimal one (reward 0.6). We deliberately initialize the policy as a sharp distribution centered on the suboptimal arm. This setup is analogous to the LLM fine-tuning challenge, where pretrained models may exhibit a bias toward solutions that are good but may not globally optimal.

We employ our risk-sensitive policy gradient algorithm to train policies with varying risk-sensitivity parameters ($\beta \geq 0$). The learning curves in Figure 3 reveal a evident divergence in performance. The

standard risk-neutral policy ($\beta = 0$) and its low-risk-sensitivity counterparts (e.g., $\beta < 4$) rapidly converge to the suboptimal reward of 0.6, becoming trapped in the local optimum by exploiting the initial policy's high-probability region. By contrast, policies with sufficient risk-seeking behavior ($\beta \geq 4$) successfully escape this trap and converge to the globally optimal reward of 1.0. The evolution of the policy distribution during training is illustrated in Figure 1.

## 3.2 THEORETICAL PERSPECTIVE

In this section, We examine the one-step policy update in a simple multi-armed bandit setting, demonstrating the fundamental advantage of the risk-sensitive objective from Eq. (5). For clarity, we assume the uniqueness of the optimal action. Our results can be generalized to settings with multiple optimal actions, which we defer to Appendix C.

**Setup and Notation** We study the $K$-armed bandit problem with action space $\mathcal{A} := \{a_1, ..., a_K\}$ and denote the unique optimal arm by $a^* \in \mathcal{A}$. We assume a bounded reward function $r : \mathcal{A} \to [0, 1]$, and consider the softmax policy $\pi_\theta$ parameterized by $\theta \in \mathbb{R}^K$: $\forall i \in [K], \quad \pi_\theta(a_i) = e^{\theta_i} / \sum_{j=1}^K e^{\theta_j}$.

Let's compare a single policy update step. Given a starting policy $\pi_\theta$, we denote the updated parameters after one step of standard policy gradient (Eq. (2)) as $\tilde{\theta}$, and after one step of risk-sensitive policy gradient (Eq. (6)) as $\tilde{\theta}^\beta$. The learning rate is assumed to be the same and omitted for simplicity.

Our first result highlights a critical flaw in the standard policy gradient: it can decrease the probability of the optimal action. This happens if a suboptimal action exists that is nonetheless better than the average, which can misdirect the update.

**Lemma 2.** *If there is an action $a_i$ with reward $r(a_i)$ such that $r(a^*) > r(a_i) > \min_j r(a_j)$, then there exist policy parameters $\theta$ for which the standard policy gradient update decreases the probability of the optimal action, i.e., $\pi_{\tilde{\theta}}(a^*) < \pi_\theta(a^*)$.*

In contrast, our next lemma states that the risk-sensitive policy gradient guarantees an improvement for the optimal action, as long as $\beta$ is sufficiently large.

**Lemma 3.** *For any policy $\pi_\theta$ and reward function $r$, there is a risk-sensitivity level $\bar{\beta}$ such that for all $\beta > \bar{\beta}$, the risk-sensitive update increases the probability of the optimal action: $\pi_{\tilde{\theta}^\beta}(a^*) > \pi_\theta(a^*)$.*

Together, Lem. 2 and Lem. 3 explain why increasing $\beta$ helps to escape from local optima in Fig. 3, and provide theoretical insights into the benefits of risk-sensitive policy gradients.

This raises a natural question: should we always use the largest possible $\beta$? The next lemma gives a negative answer: once $\beta$ exceeds a certain threshold, the policy improvement on $a^*$—while still positive—decreases as $\beta$ grows.

**Lemma 4.** *For any policy $\pi_\theta$ and reward function $r$, there is a threshold $\bar{\beta}$ such that for any $\beta_1 > \beta_2 > \bar{\beta}$, the improvement on the optimal action is smaller for the larger $\beta$: $0 < \pi_{\tilde{\theta}^{\beta_1}}(a^*) - \pi_\theta(a^*) < \pi_{\tilde{\theta}^{\beta_2}}(a^*) - \pi_\theta(a^*)$.*

This result aligns with the convergence speed shown on the right of Fig. 3, where increasing $\beta$ eventually slows down the convergence. This provides crucial guidance for tuning $\beta$ in practice: it should be large to enhance exploration, but not so large that it hinders convergence speed.

## 4 EXPERIMENTS

### 4.1 SETUP

**Training Setting** We focus on mathematical reasoning tasks and train our approach on six base models: Qwen2.5-Math-1.5B, Qwen2.5-Math-7B, Qwen2.5-7B, Qwen3-4B-Base, and LLama3.1-8B-Instruct (Yang et al., 2024a; 2025; Grattafiori et al., 2024). **We name our method RS-GRPO, which extends the GRPO algorithm with the risk-sensitive advantage function (Eq. 8).** The training framework is built upon VeRL (Sheng et al., 2024) and incorporates techniques from DAPO (Yu et al., 2025), such as dynamic sampling (filtering samples with all-0 or all-1 accuracy in each rollout) and clip-higher. We keep the shared hyperparameters identical across all comparative experiments. Full details of the training are provided in the Appendix F.

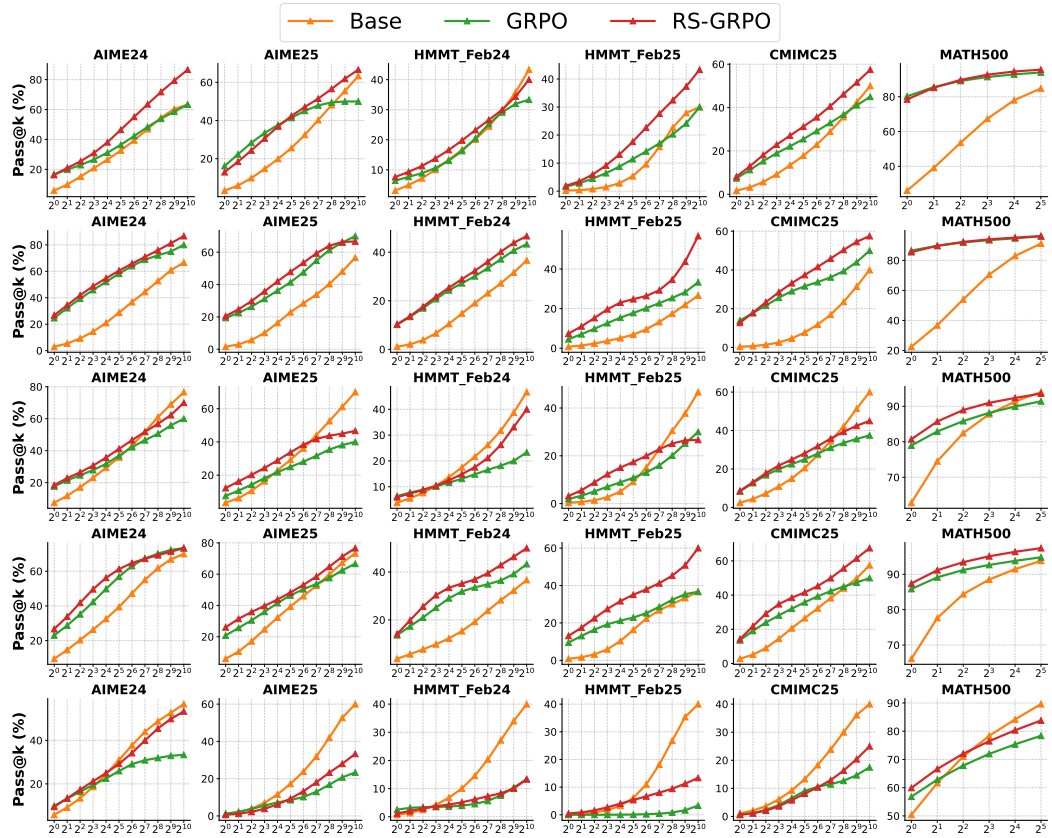

Figure 4: **Pass@k performance of RS-GRPO, GRPO, and base models.** (Qwen2.5-Math-1.5B, Qwen2.5-Math-7B, Qwen2.5-7B, Qwen3-4B-Base, and Llama3.1-8B-Instruct from top to bottom.)

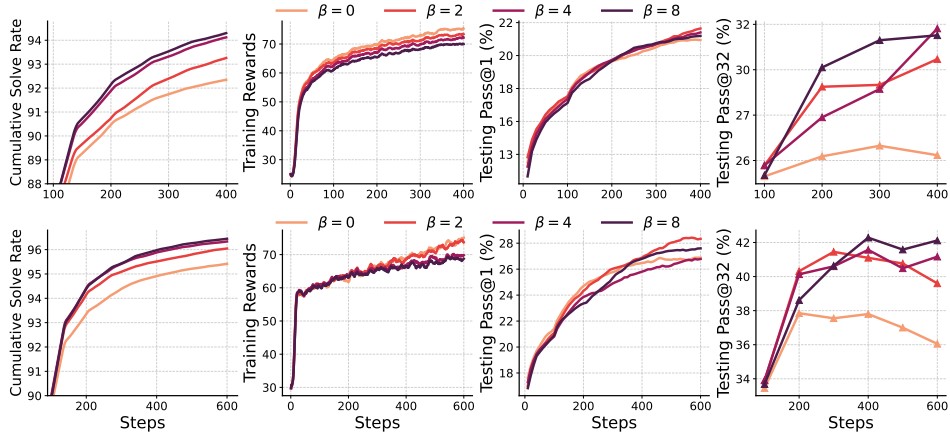

Figure 5: **Ablation Study of $\beta$ in RS-GRPO** on Qwen2.5-Math-1.5B (top) and -7B (bottom).

**Evaluation Setting** Our evaluation is conducted on six widely-used mathematical reasoning benchmarks: MATH500 (Cobbe et al., 2021), AIME24, AIME25 (Mathematical Association of America, 2025), HMMT-Feb24, HMMT-Feb25 (Harvard-MIT Mathematics Tournament, 2025), and CMIMC25 (Carnegie Mellon Informatics and Mathematics Competition, 2025). The MATH500 benchmark contains 500 problems, while the other datasets consist of 30 or 40 problems each. For most benchmarks, we generate $N = 1024$ candidate solutions per problem. However, for the larger MATH500 dataset, we use $N = 32$ to ensure the evaluation remains computationally feasible.

## 4.2 PERFORMANCE ON PASS@K EVALUATION OF RISK-SENSITIVE RL

We present the pass@$k$ performance for $k \in \{1, 2, \ldots, 1024\}$ across five LLMs and six benchmarks in Fig. 4. The results reveal that RS-GRPO consistently and significantly outperforms both the the standard GRPO baseline, showing comprehensive improvements on the pass@$k$ metric. Notably, for several models (e.g., Qwen2.5-Math-1.5B, Qwen2.5-Math-7B, and Qwen3-4B), GRPO underperforms the base model at high values of $k$ ($k > 256$). This suggests that GRPO merely sharpens the existing policy distribution rather than discovering novel solutions. In contrast, RS-GRPO surpasses the base model's performance, demonstrating its ability to expand the model's exploratory boundaries. However, for some models, such as Qwen2.5-7B and Llama3.1-8B-Instruct, RS-GRPO fails to outperform the base model at high values of $k$. We speculate this occurs when the optimal policy is prohibitively distant from the initial distribution, causing RS-GRPO to converge to a local optimum. Nonetheless, this still represents a significant improvement over GRPO.

## 4.3 DIFFERENT IMPACT OF RISK-SENSITIVE HYPERPARAMETER $\beta$

We conduct an ablation study on the risk-sensitive parameter $\beta$ to analyze its impact on training dynamics. We track several key metrics—including the cumulative solve rate on the training set (a problem is counted as solved if at least one generated response is correct; this equals the fraction of training problems solved), training reward, and test performance (pass@1 and pass@32)—for $\beta \in \{0, 2, 4, 8\}$. The case where $\beta = 0$ is equivalent to standard RL (i.e. GRPO).

Figure 5 illustrates the training dynamics for the Qwen2.5-Math-1.5B and Qwen2.5-Math-7B models. We observe that as $\beta$ increases, the cumulative solve rate on the training data improves, while the training reward grows more slowly. This result aligns with the theoretical analysis in Sec. 3.2. This slower reward growth is not necessarily a drawback, as it may indicate a regularization effect that prevents overfitting. On the test benchmarks, RS-GRPO yields substantial gains in pass@32 performance, with an improvement of approximately 5%. While pass@1 performance is maintained relative to standard RL, an appropriate choice of $\beta$ (e.g., $\beta = 2$) can lead to a 1-2% improvement, as observed on Qwen2.5-Math-7B. This suggests that balancing the objectives of optimizing for the mean reward (pass@1) and the maximum reward (pass@$k$) is crucial. **We conclude that $\beta = 2$ offers an effective trade-off, achieving strong pass@k performance while simultaneously enhancing pass@1.**

## 4.4 COMPARISON TO OTHER PASS@K OPTIMIZATION BASELINES

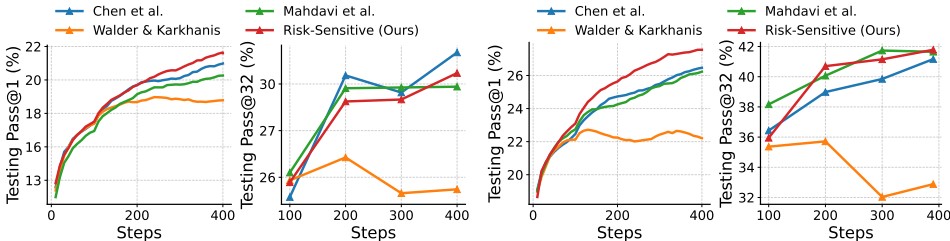

Figure 6: Training dynamics of Risk-Sensitive RL vs. other Pass@k optimization methods. (Left: Qwen2.5-Math-1.5B. Right: Qwen2.5-Math-7B).

We compare RS-GRPO with several pass@k optimization baselines (Walder & Karkhanis, 2025; Mahdavi et al., 2025; Chen et al., 2025) on the Qwen2.5-Math-1.5B and Qwen2.5-Math-7B models using the deepmath103k dataset. Figure 6 shows the training dynamics: RS-GRPO generally matches the pass@32 performance of baselines while consistently outperforming them in pass@1. We attribute this improvement to the denser advantage signals provided by our risk-sensitive objective, as discussed in related work.

Table 2 provides a more comprehensive evaluation, covering five base models and three training datasets (math12k, deepmath103k, dapo17k). While many pass@k-oriented baselines fail to improve pass@1 over GRPO, RS-GRPO achieves at least comparable Pass@1 performance and exceeds GRPO by an average of about 2% across three models (Qwen2.5-7B-Math, Qwen2.5-7B, Qwen3-4B). In addition, RS-GRPO consistently improves pass@32 over GRPO by an average of about 4%.

Table 2: **Main results on mathematical reasoning benchmarks, reporting pass@1 and pass@32 (%) for five models and three training datasets. Subscripts denote improvement over GRPO.** RS-GRPO consistently outperforms the GRPO baseline on pass@32, while maintaining or improving pass@1 accuracy. RS-GRPO also achieves a better trade-off than prior pass@k optimization methods.

| | AIME24 | AIME25 | HMMT_Feb24 | HMMT_Feb25 | CMIMC25 | MATH500 | Average |
|---|---|---|---|---|---|---|---|
| *Qwen2.5-Math-1.5B (deepmath103k): Pass@1 / Pass@32* | | | | | | | |
| Base | 5.8 / 32.6 | 3.3 / 25.6 | 3.1 / 16.7 | 0.2 / 5.3 | 1.7 / 17.8 | 26.0 / 84.8 | 6.7 / 30.5 |
| GRPO | 16.6 / 36.4 | 16.2 / 41.4 | 6.3 / 16.3 | 1.6 / 11.4 | 7.4 / 25.4 | **80.2** / 94.0 | **21.4** / 37.5 |
| Walder & Karkhanis (2025) | 13.7 / 34.5 | 10.0 / 38.2 | 5.9 / 15.7 | 1.5 / 13.7 | 4.3 / 25.0 | 64.7 / 92.6 | 16.7 / 36.6 |
| Mahdavi et al. (2025) | 15.9 / 45.0 | 15.2 / **43.4** | 6.9 / **22.7** | 1.2 / 15.5 | 7.4 / **31.6** | 75.5 / 95.4 | 20.4 / **42.3** |
| Chen et al. (2025) | 16.2 / 44.1 | 14.6 / 41.9 | 5.6 / 20.9 | **1.9** / 16.4 | **8.0** / 29.3 | 79.1 / 94.3 | 20.9 / 41.2 |
| **RS-GRPO** | **16.7**$_{(+0.1)}$ / **45.1**$_{(+8.7)}$ | **16.9**$_{(+0.7)}$ / 42.8$_{(+1.4)}$ | **7.2**$_{(+0.9)}$ / 19.9$_{(+3.6)}$ | 1.7$_{(+0.1)}$ / **17.6**$_{(+6.2)}$ | 7.2$_{(-0.2)}$ / 30.7$_{(+5.3)}$ | 78.1$_{(-2.1)}$ / **95.6**$_{(+1.6)}$ | 21.3$_{(-0.1)}$ / 42.0$_{(+4.5)}$ |
| *Qwen2.5-Math-7B (deepmath103k): Pass@1 / Pass@32* | | | | | | | |
| Base | 2.9 / 28.8 | 1.6 / 22.9 | 1.1 / 14.8 | 0.7 / 6.9 | 0.4 / 7.7 | 22.5 / 91.2 | 4.9 / 28.7 |
| GRPO | 25.7 / 58.0 | 19.3 / 41.2 | 10.2 / 26.2 | 7.6 / 26.1 | 11.2 / 24.1 | 85.4 / 96.0 | 26.6 / 45.3 |
| Mahdavi et al. (2025) | 24.6 / 61.8 | 19.0 / **48.6** | 9.4 / **36.9** | 7.6 / 23.6 | 10.9 / 37.4 | 85.8 / **97.8** | 26.2 / **51.0** |
| Chen et al. (2025) | 26.1 / **62.3** | **21.1** / 47.1 | 8.2 / 30.9 | 5.8 / 24.3 | 10.9 / 36.2 | 85.6 / 97.8 | 26.3 / 49.8 |
| **RS-GRPO** | **30.2**$_{(+4.5)}$ / 60.0$_{(+2.0)}$ | 20.8$_{(+1.5)}$ / 45.1$_{(+3.9)}$ | **11.6**$_{(+1.4)}$ / 29.4$_{(+3.2)}$ | **8.0**$_{(+0.4)}$ / **26.8**$_{(+0.7)}$ | **14.7**$_{(+3.5)}$ / 32.8$_{(+8.7)}$ | **86.0**$_{(+0.6)}$ / 95.8$_{(-0.2)}$ | **28.6**$_{(+2.0)}$ / 48.3$_{(+3.0)}$ |
| *Qwen2.5-Math-7B (dapo17k): Pass@1 / Pass@32* | | | | | | | |
| Base | 2.9 / 28.8 | 1.6 / 22.9 | 1.1 / 14.8 | 0.7 / 6.9 | 0.4 / 7.7 | 22.5 / 91.2 | 4.9 / 28.7 |
| GRPO | **32.1** / 61.0 | **18.7** / 37.6 | 12.9 / 23.5 | 3.0 / 13.8 | 2.7 / 11.8 | 77.8 / 92.2 | 24.5 / 40.0 |
| Chen et al. (2025) | 28.7 / 67.4 | 17.6 / 44.6 | 11.8 / 25.8 | **4.0** / 19.3 | 5.2 / **22.8** | 79.0 / **94.8** | 24.4 / **45.8** |
| Mahdavi et al. (2025) | 27.8 / **68.9** | 15.6 / **46.3** | 11.5 / 24.7 | 3.8 / **19.6** | 3.2 / 15.6 | 75.5 / 93.6 | 22.9 / 44.8 |
| **RS-GRPO** | **34.2**$_{(+2.1)}$ / 65.8$_{(+4.8)}$ | **18.7** / 40.7$_{(+3.1)}$ | **16.4**$_{(+3.5)}$ / 28.3$_{(+4.8)}$ | 3.5$_{(+0.5)}$ / 16.8$_{(+3.0)}$ | **5.4**$_{(+2.7)}$ / 20.4$_{(+8.6)}$ | **80.4**$_{(+2.6)}$ / 94.8$_{(+2.6)}$ | **26.4**$_{(+1.9)}$ / 44.5$_{(+4.5)}$ |
| *Qwen2.5-Math-7B (math12k): Pass@1 / Pass@32* | | | | | | | |
| Base | 2.9 / 28.8 | 1.6 / 22.9 | 1.1 / 14.8 | 0.7 / 6.9 | 0.4 / 7.7 | 22.5 / 91.2 | 4.9 / 28.7 |
| GRPO | **34.0** / 58.3 | 13.9 / 36.5 | **10.1** / 23.9 | 1.2 / 14.2 | 4.8 / 23.9 | 78.4 / 94.2 | 23.7 / 41.8 |
| **RS-GRPO** | 33.1$_{(-0.9)}$ / **59.4**$_{(+1.1)}$ | **16.7**$_{(+2.8)}$ / **37.6**$_{(+1.1)}$ | 10.0$_{(-0.1)}$ / **27.2**$_{(+3.3)}$ | **1.4**$_{(+0.2)}$ / **14.5**$_{(+0.3)}$ | **5.9**$_{(+1.1)}$ / **26.8**$_{(+2.9)}$ | **81.5**$_{(+3.1)}$ / **94.8**$_{(+0.6)}$ | **24.8**$_{(+1.1)}$ / **43.4**$_{(+1.6)}$ |
| *Qwen2.5-7B (math12k): Pass@1 / Pass@32* | | | | | | | |
| Base | 7.4 / 35.8 | 3.4 / 29.2 | 3.8 / 17.5 | 0.4 / 9.1 | 2.4 / 20.5 | 62.8 / 94.0 | 13.4 / 34.4 |
| GRPO | 17.7 / 36.9 | 7.5 / 25.0 | 6.4 / 13.1 | 1.9 / 10.8 | 8.4 / 25.0 | 79.0 / 91.4 | 20.2 / 33.7 |
| **RS-GRPO** | **18.5**$_{(+0.8)}$ / **41.1**$_{(+4.2)}$ | **12.2**$_{(+4.7)}$ / **33.7**$_{(+8.7)}$ | **6.0**$_{(-0.4)}$ / **14.9**$_{(+1.8)}$ | **3.2**$_{(+1.3)}$ / **17.4**$_{(+6.6)}$ | **8.6**$_{(+0.2)}$ / **28.1**$_{(+3.1)}$ | **80.7**$_{(+1.7)}$ / **93.6**$_{(+2.2)}$ | **21.5**$_{(+1.3)}$ / **38.1**$_{(+4.4)}$ |
| *Qwen3-4B-Base (math12k): Pass@1 / Pass@32* | | | | | | | |
| Base | 9.5 / 39.4 | 5.9 / 39.3 | 3.7 / 15.3 | 0.8 / 16.3 | 2.7 / 26.5 | 66.1 / 93.8 | 14.8 / 38.4 |
| GRPO | 23.0 / 56.9 | 20.8 / 46.5 | 13.6 / 31.9 | 9.5 / 22.9 | 13.6 / 35.8 | 85.9 / 94.8 | 27.7 / 48.1 |
| **RS-GRPO** | **26.7**$_{(+3.7)}$ / **61.2**$_{(+4.3)}$ | **26.1**$_{(+5.3)}$ / **48.3**$_{(+1.8)}$ | **14.1**$_{(+0.5)}$ / **35.2**$_{(+3.3)}$ | **13.1**$_{(+3.6)}$ / **35.1**$_{(+12.2)}$ | **14.2**$_{(+0.6)}$ / **41.6**$_{(+5.8)}$ | **87.4**$_{(+1.5)}$ / **97.4**$_{(+2.6)}$ | **30.3**$_{(+2.6)}$ / **53.1**$_{(+5.0)}$ |
| *Llama-3.1-8B-Instruct (math12k): Pass@1 / Pass@32* | | | | | | | |
| Base | 5.8 / 31.0 | 1.1 / 17.2 | 0.7 / 10.1 | 0.2 / 6.1 | 1.1 / 13.3 | 50.4 / 89.6 | 9.9 / 27.9 |
| GRPO | **9.9** / 25.9 | **1.2** / 8.5 | **2.5** / 3.9 | 0.0 / 0.1 | 0.6 / 9.1 | 56.8 / 78.4 | 11.8 / 21.0 |
| **RS-GRPO** | 9.4$_{(-0.5)}$ / **29.2**$_{(+3.3)}$ | 0.6$_{(-0.6)}$ / **9.3**$_{(+0.8)}$ | 1.2$_{(-1.3)}$ / **5.1**$_{(+1.2)}$ | **0.5**$_{(+0.5)}$ / **5.4**$_{(+5.3)}$ | **0.6** / 8.1$_{(-1.0)}$ | **59.9**$_{(+3.1)}$ / **83.8**$_{(+5.4)}$ | **12.0**$_{(+0.2)}$ / **23.5**$_{(+2.5)}$ |

We observe that the approach of Walder & Karkhanis (2025) performs unsatisfactorily, mainly because its advantage estimates remain strictly positive (see Appendix D). The absence of negative advantages causes rapid entropy collapse and poor training performance, consistent with prior findings on the importance of negative signals (Zhu et al., 2025).

## 4.5 ANALYSIS OF PASS@K IMPROVEMENT

We analyze how risk-sensitive RL enhances both pass@k and pass@1 performance using the Qwen2.5-Math-7B model trained on the deepmath103k dataset. For each problem in five benchmarks—AIME24, AIME25, HMMT_Feb24, HMMT_Feb25, and CMIMC25—we sample 1024 solutions, each corresponding to a single final answer. The left of Figure 7 illustrates the unique answers ratio. We observe that after RS-GRPO training, the number of unique answers shows a significant increase compared to that of GRPO. This indicates that risk-sensitive RL enhances the diversity of reasoning paths.

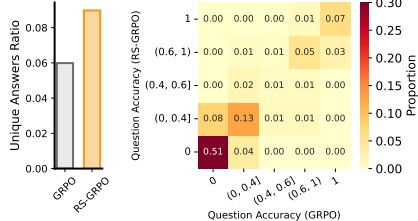

Figure 7: **Left:** RS-GRPO finds more unique solutions. **Right:** Accuracy transition map from GRPO to RS-GRPO.

The heatmap in Figure 7 provides a detailed view of the prompt accuracy transitions from GRPO to RS-GRPO. We observe that 8% of prompts with an accuracy of 0 under GRPO achieve an accuracy in the (0, 0.4] range with RS-GRPO, while only 4% show the opposite change. This shift is the primary contributor to the improved pass@k performance. Simultaneously, 3% of prompts with an accuracy of 1 are shifted to the (0.6, 1) range, while only 1% move in the opposite direction. This performance trade-off explains why pass@1 improvements are more modest than gains in pass@k: while RS-GRPO can solve more problems than GRPO, it occasionally makes errors on simpler ones.

## 4.6 PERFORMANCE OF RISK-SENSITIVE RL ON OOD TASKS.

We directly evaluated Qwen2.5-Math7B models trained on the deepmath103 dataset (whose math performance was already reported in Table 2) on GPQA-Diamond (Rein et al., 2024), which covers biology, physics, and chemistry. The results show that RS-GRPO consistently outperforms GRPO from pass@1 to pass@32, aligning with our findings on math tasks.

Table 3: Pass@k comparison on GPQA-Diamond.

| Model | Pass@1 (%) | Pass@2 (%) | Pass@4 (%) | Pass@8 (%) | Pass@16 (%) | Pass@32 (%) |
|---|---|---|---|---|---|---|
| Base | 4.4 | 8.4 | 15.5 | 27.3 | 44.5 | 64.6 |
| GRPO | 41.2 | 52.2 | 62.8 | 71.6 | 78.5 | 84.8 |
| RS-GRPO (ours) | **41.8** | **54.1** | **65.5** | **75.2** | **83.1** | **88.9** |

Table 4: Pass@k comparison on LiveCodeBenchv5.

| | Pass@1 (%) | Pass@2 (%) | Pass@4 (%) | Pass@8 (%) | Pass@16 (%) | Pass@32 (%) |
|---|---|---|---|---|---|---|
| Base | 22.6 | 27.4 | 31.8 | 35.6 | 39.0 | 42.1 |
| GRPO | 28.5 | 30.4 | 31.7 | 32.7 | 33.6 | 34.3 |
| RS-GRPO (ours) | **29.7** | **32.5** | **34.6** | **36.2** | **37.7** | **39.1** |

## 4.7 PERFORMANCE OF RS-GRPO ON CODING TASKS.

We conducted additional code generation experiments with Qwen2.5-7B-Instruct-1M on the HuggingFace dataset `ganler/code-r1-12k`, and evaluated on LiveCodeBench v5 (Jain et al., 2025) (880 questions in total). We set the maximum response length to 4k tokens and keep all other hyperparameters identical to Table 3. The results show that RS-GRPO outperforms GRPO on coding tasks, with a 1.2% improvement in pass@1 and a 4.5% improvement in pass@8. While the base model's pass@8 exceeds GRPO's trained result, RS-GRPO's pass@8 still surpasses the base model.

## 4.8 ABLATION KL

Our main experiments omit KL regularization, following the prevalent RLVR practice. To verify that RS-GRPO's exploration benefits persist under explicit KL control, we additionally compare RS-GRPO+KL with GRPO+KL by tracking the evolution of the average Pass@$\{1, \ldots, 32\}$ across five benchmarks (AIME24, AIME25, CMIMC25, HMMT_Feb24, HMMT_Feb25) during training. As shown in Figure 8, we trained Qwen2.5-Math-7B on the dapo17k dataset with KL regularization (coefficient 0.001). RS-GRPO+KL consistently surpasses GRPO+KL from Pass@1 to Pass@32, although the KL term slightly lowers the overall performance ceiling.

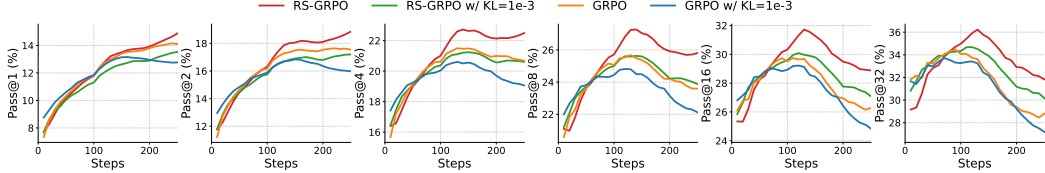

Figure 8: Ablation KL on RS-GRPO.

## 5 CONCLUSION

In this paper, we aim to address the exploration dilemma in fine-tuning large language models with reinforcement learning, where existing methods often improve pass@1 accuracy at the expense of solution diversity, leading to stagnation or even degradation in pass@k performance. We argue that this arises from the failure of standard RL algorithms to escape the local optima defined by the sharply-peaked initial policy of pretrained models. To overcome this, we introduce a risk-sensitive reinforcement learning framework, instantiated as the RS-GRPO algorithm. By optimizing a risk-seeking objective, our method encourages the policy to explore under-explored regions of the solution space, discovering novel reasoning paths. Our experiments on mathematical reasoning benchmarks demonstrate that RS-GRPO significantly improves pass@k performance while maintaining or improving pass@1 accuracy, achieving a more favorable trade-off than prior methods. Future work could explore the application of risk-sensitive objectives to other generative modeling domains and investigate their interplay with other exploration techniques (see Appendix A for limitations).

## REPRODUCIBILITY STATEMENT

We are committed to ensuring the reproducibility of our research. Our novel algorithm, RS-GRPO, is detailed in Section 2, with theoretical claims and their complete proofs provided in Appendix C. All datasets used for training and evaluation are publicly available, with specific sources and versions documented in Appendix F. This appendix also contains a comprehensive description of our experimental setup, including all hyperparameter settings (Table 5) and training configurations. To facilitate direct replication of our results, we provide the key source code as part of the supplementary material.

## ACKNOWLEDGEMENTS

This work was supported in part by the National Natural Science Foundation of China under Grant 62192751, in part by the Key R&D Project of China under Grant 2017YFC0704100, in part by the 111 International Collaboration Program of China under Grant B25027, in part by BNRist Program under Grant BNR2019TD01009, in part by the National Innovation Center of High Speed Train R&D Project under Grant CX/KJ-2020-0006, in part by the InnoHK Initiative, The Government of HKSAR, and in part by the Laboratory for AI-Powered Financial Technologies.

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

## A    LIMITATIONS

A limitation of our current work is that all experiments were conducted with a fixed risk-seeking parameter $\beta$. A natural extension is to dynamically adjust $\beta$ during training to better balance exploration and exploitation. We experimented with several heuristics, including:

- Initiating training with a high $\beta$ value, followed by a linear or cosine decay schedule after an initial training period.
- Beginning with a large $\beta$ and subsequently switching to standard mean-reward optimization after an initial phase.
- Employing an adaptive $\beta$ based on prompt difficulty, assigning larger values to harder prompts and smaller values to easier ones.

However, none of these strategies yielded superior pass@1 performance compared to training with a fixed, well-chosen $\beta$ (i.e., $\beta = 2$). Devising an optimal dynamic strategy to balance exploration and exploitation remains a challenging open problem.

Another limitation is that our methods are tailored to bandit settings. Extending them to general Markov Decision Processes (MDPs) is non-trivial, because credit assignment in MDPs requires learning a value function, and a principled design for risk-sensitive value functions remains challenging (Jiang et al., 2024; Delétang et al., 2021). Fortunately, most RLVR tasks—including multi-turn ones such as coding and search agents—only reveal a reward at the end of generation, so they can be naturally modeled as bandit problems and handled by our approach. In short, whenever a GRPO-style algorithm is applicable, our risk-sensitive RL can be directly deployed.

## B    THE USE OF LARGE LANGUAGE MODELS (LLMS)

We utilized a large language model to enhance the quality of this paper. Its applications included refining the manuscript for clarity, conciseness, and grammatical correctness.

## C    MISSING PROOFS

### C.1    PROOF FOR THEOREM 1

*Proof.* The proof relies on the log-derivative trick ($\nabla_\theta \pi_\theta = \pi_\theta \nabla_\theta \log \pi_\theta$). The gradient of $\mathcal{J}_x = \frac{1}{\beta} \log \mathbb{E}_{y \sim \pi_\theta}[e^{\beta r(y)}]$ is:

$$\nabla_\theta \mathcal{J}_x = \frac{1}{\beta} \frac{\nabla_\theta \mathbb{E}_{y \sim \pi_\theta}[e^{\beta r(y)}]}{\mathbb{E}_{y \sim \pi_\theta}[e^{\beta r(y)}]} = \frac{1}{\beta} \frac{\mathbb{E}_{y \sim \pi_\theta}[e^{\beta r(y)} \nabla_\theta \log \pi_\theta(y|x)]}{\mathbb{E}_{y' \sim \pi_\theta}[e^{\beta r(y')}]} \qquad \text{(Log-derivative trick)}$$

$$= \mathbb{E}_{y \sim \pi_\theta}\left[ \frac{e^{\beta r(y)}}{\beta \cdot \mathbb{E}_{y' \sim \pi_\theta}[e^{\beta r(y')}]} \nabla_\theta \log \pi_\theta(y|x) \right].$$

Here, $y'$ is a dummy variable for the inner expectation. Subtracting the baseline $1/\beta$ from the advantage term gives the final form in Eq. (7), which is an unbiased estimator with reduced variance.    □

### C.2    THEORETICAL ANALYSIS OF RISK-SENSITIVE POLICY GRADIENT

In this section, we provide the formal version of lemmas in Sec. 3.2, and the detailed proofs.

We study the general cases without restricting the uniqueness of the optimal arm. Recall that we consider the bandit setting with $K$ actions $\mathcal{A} := \{a_1, ..., a_K\}$. We will use $\mathcal{I}^* := \{i \in [K] | r(a_i) = \max_{j \in [K]} r(a_j)\}$ to refer to the collection of indices of all optimal actions. With a bit abuse of notation, we denote $\pi(\mathcal{I}^*) := \sum_{i \in \mathcal{I}^*} \pi(a_i)$ to be the total mass of $\pi$ on optimal actions.

We consider the softmax policy $\pi_\theta$ parameterized by $\theta := [\theta_1, ..., \theta_K] \in \mathbb{R}^K$:

$$\forall i \in [K], \quad \pi_\theta(a_i) = \frac{e^{\theta_i}}{\sum_{j \in [K]} e^{\theta_j}}.$$

Following the notation in Sec. 3.2, we denote $\tilde{\theta} := [\tilde{\theta}_1, ..., \tilde{\theta}_K] \in \mathbb{R}^K$ and $\tilde{\theta}^\beta := [\tilde{\theta}_1^\beta, ..., \tilde{\theta}_K^\beta] \in \mathbb{R}^K$ to be the parameters after performing one-step standard PG and risk-sensitive PG on $\theta$, respectively. Combining with the policy gradient theorem for softmax policy in (Mei et al., 2020), as implied by Eq. (2) and Eq. (6), elementwisely, the updates of the policy parameters follow:

$$\forall i \in [K], \quad \tilde{\theta}_i \leftarrow \theta_i + \alpha \pi_\theta(a_i) A^{\pi_\theta}(a_i), \tag{9}$$

$$\tilde{\theta}_i \leftarrow \theta_i^\beta + \alpha \pi_\theta(a_i) A_\beta^{\pi_\theta}(a_i), \tag{10}$$

where $\alpha > 0$ denotes an arbitrary and shared learning rate.

For simplicity, we use $\pi_i := \pi_\theta(a_i)$ and $A_i := A^{\pi_\theta}(a_i)$ as short notes of policy and advantage values regarding $\theta$, and use $\tilde{\pi}_i := \pi_{\tilde{\theta}}(a_i)$ and $\tilde{\pi}_i^\beta := \pi_{\tilde{\theta}^\beta}(a_i)$ as the short note of policy value w.r.t. the parameters after being updated. Similarly, $\pi_{\mathcal{I}^*}, \tilde{\pi}_{\mathcal{I}^*}$ and $\tilde{\pi}_{\mathcal{I}^*}^\beta$ denote the total policy density assigned to the set of optimal actions.

By Eq. (9), the dynamics of $\tilde{\pi}$ and $\tilde{\pi}^\beta$ follow:

$$\tilde{\pi}_i = \frac{e^{\theta_i + \alpha \pi_i A_i}}{\sum_j e^{\theta_j + \alpha \pi_j A_j}} = \frac{e^{\theta_i}}{\sum_j e^{\theta_j + \alpha \pi_j A_j - \alpha \pi_i A_i}} = \frac{\pi_i}{\sum_j \pi_j e^{\alpha \pi_j A_j - \alpha \pi_i A_i}}. \tag{11}$$

**Remark 5.** *Note that $\min_{i \in [K]} r(a_i) = \max_{i \in [K]} r(a_j)$ is a trivial case where every action is optimal. We only focus on cases where $\min_{i \in [K]} r(a_i) < \max_{i \in [K]} r(a_j)$.*

**Lemma 6.** *[Formal Version of Lem. 2] As long as $\exists i' \in [K]$ satisfying $\max_i r(a_i) > r(a_{i'}) > \min_i r(a_i)$, there exists $\theta$ (or equivalently, $\pi_\theta$), s.t., $\tilde{\pi}_{\mathcal{I}^*} < \pi_{\mathcal{I}^*}$, or even, $\tilde{\pi}_i < \pi_i$ for any $i \in \mathcal{I}^*$ and any learning rate $\alpha > 0$.*

*Proof.* The proof is by construction. By Eq. (11), $\tilde{\pi}_i < \pi_i$ as long as $\sum_j \pi_j e^{\alpha \pi_j A_j - \alpha \pi_i A_i} > 1$. By the convexity of exponential function,

$$\sum_j \pi_j e^{\alpha \pi_j A_j - \alpha \pi_i A_i} \geq e^{\alpha(\sum_j \pi_j^2 A_j - \pi_i A_i)},$$

and all we need to do is to construct a $\pi_\theta$ s.t. the RHS above is larger than 1.

For convenience, we denote $i^- := \arg\min_{i \in [K]} r(a_i)$ to be (one of) the worst action(s), and denote $i'$ to be (one of) the second optimal actions such that $r(a_{i'}) = \max_{i \in [K] \setminus \mathcal{I}^*} r(a_i)$. Besides, we denote $r_{\max} := \max_i r(a_i)$ and $r_{\min} := \min_i r(a_i)$ as the maximal and minimal policy values, respectively. Note that,

$$A_{i'} = r(a_{i'}) - \sum_{i \in [K]} \pi_i r(a_i) > \pi_{i^-}(r(a_{i'}) - r_{\min}) - \pi_{\mathcal{I}^*}(r_{\max} - r(a_{i'})).$$

Consider an arbitrary policy satisfying the following constraint:

$$0 < \pi_{\mathcal{I}^*} < \frac{1}{2} \cdot \frac{r(a_{i'}) - r_{\min}}{r_{\max} - r(a_{i'})} \pi_{i^-}, \tag{12}$$

which implies $A_{i'} > \frac{1}{2} \pi_{i^-}(r(a_{i'}) - r_{\min})$.

Since $A_j \in [-1, 1]$, under the constraints by Eq. (12) and that $\forall i, \pi_i > 0$ and $\sum_i \pi_i = 1$, as long as $\pi_{i'} \geq \frac{1}{2}$, for any optimal action index $i^* \in \mathcal{I}^*$, we have:

$$\sum_j \pi_j^2 A_j - \pi_{i^*} A_{i^*} \geq \pi_{i'}^2 A_{i'} - \pi_{i^*} - \sum_{i \neq i'} \pi_j^2 \geq \frac{1}{8} \pi_{i^-}(r(a_{i'}) - r_{\min}) - \pi_{i^*} - \sum_{j \neq i'} \pi_j^2. \tag{13}$$

We consider the following constraints, which are feasible when $\pi_{i'}$ is large enough and $\pi_{i^-}$ is appropriately chosen:

$$\pi_{i^-}^2 < \frac{\pi_{i^-}}{16}(r(a_{i'}) - r_{\min}),$$

$$\pi_{i^*} + \sum_{j \neq i', j \neq i^-} \pi_j^2 < \frac{\pi_{i^-}}{16}(r(a_{i'}) - r_{\min}).$$

In this case, we can conclude $\sum_j \pi_j^2 A_j - \pi_{i^*} A_{i^*} > 0$ from Eq. (13). This implies $\tilde{\pi}_{i^*} < \pi_{i^*}$ for any optimal action $a_{i^*}$, and therefore,

$$\tilde{\pi}_{\mathcal{I}^*} < \pi_{\mathcal{I}^*}.$$

$\square$

**Lemma 7.** *[Formal Version of Lem. 3] For any given $r$ and $\theta$, consider the risk-sensitive update (Eq. (6) or Eq. (10)), there always exists $\bar{\beta}$, for any $\beta > \bar{\beta}$ and $\alpha > 0$, we have $\tilde{\pi}_{\mathcal{I}^*}^{\beta} > \pi_{\mathcal{I}^*}$.*

*Proof.* In the risk sensitive setting, recall the advantage function takes

$$A_{\beta}^{\pi_\theta} := \frac{1}{\beta}\Big(\frac{e^{\beta r(a_i)}}{\mathbb{E}_{a_j \sim \pi_\theta}[e^{\beta r(a_j)}]} - 1\Big)$$

For convenience, we use $A_{\beta,i} := A_{\beta}^{\pi_\theta}(a_i)$ as a short note.

By Eq. (10), the risk-sensitive policy gradient yields:

$$\tilde{\pi}_i = \frac{\pi_i}{\sum_j \pi_j e^{\alpha(\pi_j A_{\beta,j} - \pi_i A_{\beta,i})}} = \frac{\pi_i e^{\alpha \pi_i A_{\beta,i}}}{Z}. \tag{14}$$

Here $Z := \sum_j \pi_j e^{\alpha \pi_j A_{\beta,j}}$ denotes a normalization term independent of $i$.

Now, let's denote $i'$ to be the second optimal action satisfying $r(a_{i'}) = \max_{i \in [K] \setminus \mathcal{I}^*} r(a_i)$ and denote $\Delta := \max_i r(a_i) - r(a_{i'}) > 0$ to be its value gap.

Easy to see that, for any $i \in \mathcal{I}^*$, $A_{\beta,i^*} > 0$, while for any $i \notin \mathcal{I}^*$,

$$A_{\beta,i} = \frac{1}{\beta}\Big(\frac{1}{\mathbb{E}_{a_j \sim \pi_\theta}[e^{\beta r(a_j) - \beta r(a_i)}]} - 1\Big) \leq \frac{1}{\beta}\Big(\frac{1}{\pi_{\mathcal{I}^*} e^{\beta \Delta}} - 1\Big).$$

Therefore, as long as $\beta \geq \frac{1}{\Delta} \log \frac{1}{\pi_{\mathcal{I}^*}}$, we have $A_{\beta,i} < 0$, which implies

$$\forall i \in \mathcal{I}^*, \quad \pi_i e^{\alpha \pi_i A_{\beta,i}} > \pi_i,$$
$$\forall i \notin \mathcal{I}^*, \quad \pi_i e^{\alpha \pi_i A_{\beta,i}} < \pi_i.$$

By Eq. (14), we must have $\tilde{\pi}_{\mathcal{I}^*}^{\beta} > \pi_{\mathcal{I}^*}$. $\square$

**Lemma 8.** *[Formal Version of Lem. 4] For any given $r$ and $\theta$, there exists $\bar{\beta}$, s.t., for any $\beta_1 > \beta_2 > \bar{\beta}$, $0 < \tilde{\pi}_{\mathcal{I}^*}^{\beta_1} - \pi_{\mathcal{I}^*} < \tilde{\pi}_{\mathcal{I}^*}^{\beta_2} - \pi_{\mathcal{I}^*}$ for any fixed learning rate $\alpha > 0$.*

*Proof.* We view $A_{\beta,i} := A_{\beta}^{\pi_\theta}(a_i)$ as a continuous function in $\beta$:

$$A_{\beta,i} := \frac{1}{\beta}\Big(\frac{e^{\beta r(a_i)}}{\mathbb{E}_{a_j \sim \pi_\theta}[e^{\beta r(a_j)}]} - 1\Big) = \frac{1}{\beta}\Big(\frac{1}{\mathbb{E}_{a_j \sim \pi_\theta}[e^{\beta \Delta_{j,i}}]} - 1\Big),$$

where we use $\Delta_{j,i} := r(a_j) - r(a_i)$ as a short note. By taking the derivative w.r.t. $\beta$, we have:

$$A'_{\beta,i} = -\frac{1}{\beta^2}\Big(\frac{1}{\mathbb{E}_{a_j \sim \pi_\theta}[e^{\beta \Delta_{j,i}}]} - 1\Big) - \frac{1}{\beta}\frac{\mathbb{E}_{a_j \sim \pi_\theta}[\Delta_{j,i} e^{\beta \Delta_{j,i}}]}{\mathbb{E}_{a_j \sim \pi_\theta}^2[e^{\beta \Delta_{j,i}}]}$$

$$= \frac{1}{\beta^2 \mathbb{E}_{a_j \sim \pi_\theta}^2[e^{\beta \Delta_{j,i}}]}\Big(\mathbb{E}_{a_j \sim \pi_\theta}^2[e^{\beta \Delta_{j,i}}] - \mathbb{E}_{a_j \sim \pi_\theta}[(1 + \beta \Delta_{j,i})e^{\beta \Delta_{j,i}}]\Big).$$

We first check $A'_{\beta,i}$ for $i \in \mathcal{I}^*$. Since $\Delta_{j,i} \leq 0$ for any $j$, we have:

$$\mathbb{E}_{a_j \sim \pi_\theta}^2[e^{\beta \Delta_{j,i}}] - \mathbb{E}_{a_j \sim \pi_\theta}[(1 + \beta \Delta_{j,i})e^{\beta \Delta_{j,i}}]$$

$$= (\pi_{\mathcal{I}^*} + \sum_{j: \Delta_{j,i} < 0} \pi_j e^{\beta \Delta_{j,i}})^2 - \sum_{j: \Delta_{j,i} < 0} \pi_j(1 + \beta \Delta_{j,i})e^{\beta \Delta_{j,i}} - \pi_{\mathcal{I}^*}$$

$$\leq \pi_{\mathcal{I}^*}^2 + 2\pi_{\mathcal{I}^*} \sum_{j: \Delta_{j,i} < 0} \pi_j e^{\beta \Delta_{j,i}} - \pi_{\mathcal{I}^*} + (\sum_{j: \Delta_{j,i} < 0} \pi_j e^{\beta \Delta_{j,i}})^2 + \beta \sum_{j: \Delta_{j,i} < 0} \pi_j |\Delta_{j,i}| e^{\beta \Delta_{j,i}}.$$

Now, we denote $\beta'$ to be the minimal value, s.t., for all $\beta > \beta'$,

$$\sum_{j:\Delta_{j,i}<0} \pi_j e^{\beta\Delta_{j,i}} \leq \frac{1-\pi_{\mathcal{I}^*}}{6}, \tag{15}$$

and we denote $\beta''$ to be the minimal value, s.t., for all $\beta > \beta''$,

$$(\sum_{j:\Delta_{j,i}<0} \pi_j e^{\beta\Delta_{j,i}})^2 + \beta \sum_{j:\Delta_{j,i}<0} \pi_j |\Delta_{j,i}| e^{\beta\Delta_{j,i}} \leq \frac{\pi_{\mathcal{I}^*} - \pi_{\mathcal{I}^*}^2}{3}. \tag{16}$$

Since the RHS of both Eq. (15) and Eq. (16) are independent w.r.t. $\beta$, such a $\beta'$ and $\beta''$ always exists. Then, for any $\beta > \max\{\beta', \beta''\}$, we have:

$$\mathbb{E}_{a_j\sim\pi_\theta}^2[e^{\beta\Delta_{j,i}}] - \mathbb{E}_{a_j\sim\pi_\theta}[(1+\beta\Delta_{j,i})e^{\beta\Delta_{j,i}}] \leq \frac{\pi_{\mathcal{I}^*}^2 - \pi_{\mathcal{I}^*}}{3} < 0,$$

which implies that, although $A_{\beta,i^*} > 0$, it decreases as $\beta$ increases.

Secondly, we check $A'_{\beta,i}$ for all the other $i \notin \mathcal{I}^*$. As we discussed in the proof of Lem. 7, when $\beta \geq \frac{1}{\Delta}\log\frac{1}{\pi_{\mathcal{I}^*}}$, we have $A_{\beta,i} < 0$ for all $i \neq i^*$. Note that,

$$\mathbb{E}_{a_j\sim\pi}^2[e^{\beta\Delta_{j,i}}] - \mathbb{E}_{a_j\sim\pi_\theta}[(1+\beta\Delta_{j,i})e^{\beta\Delta_{j,i}}]$$

$$\geq (\sum_{j:\Delta_{j,i}=0} \pi_j + \sum_{j:\Delta_{j,i}>0} \pi_j e^{\beta\Delta_{j,i}})^2$$

$$\text{(Dropped positive terms } \sum_{j:\Delta_{j,i}<0} \pi_i e^{\beta\Delta_{j,i}} \text{ in } \mathbb{E}_{a_j\sim\pi}^2[e^{\beta\Delta_{j,i}}])$$

$$- \sum_{j:\Delta_{j,i}=0} \pi_j - \sum_{j:\Delta_{j,i}>0} \pi_j(1+\beta\Delta_{j,i})e^{\beta\Delta_{j,i}} - \sum_{j:\Delta_{j,i}<0} \pi_j(1+\beta\Delta_{j,i})e^{\beta\Delta_{j,i}}$$

$$\geq \underbrace{(\sum_{j:\Delta_{j,i}=0} \pi_i)^2 - \sum_{j:\Delta_{j,i}=0} \pi_i}_{p_1} + \underbrace{\sum_{j:\Delta_{j,i}>0} \pi_j^2 e^{2\beta\Delta_{j,i}} - \sum_{j:\Delta_{j,i}>0} \pi_j(1+\beta\Delta_{j,i})e^{\beta\Delta_{j,i}}}_{p_2} - \underbrace{\sum_{j:\Delta_{j,i}<0} \pi_j e^{\beta\Delta_{j,i}}}_{p_3}.$$

$$(a^2 + b^2 \leq (a+b)^2 \text{ for } a,b > 0)$$

As we can see, $p_1$ is negative but fixed; for $p_3$, consider $\beta^\dagger := \max_{j:\Delta_{j,i}<0}\frac{1}{|\Delta_{j,i}|}$, we know $0 < p_3 \leq 1$ as long as $\beta \geq \beta^\dagger$. Then, we check $p_2$, which is dominated by $e^{2\beta\Delta_{j,i}}$. There exists $\beta^{\dagger\dagger}$, s.t., $p_2 > |p_1| + 1 > |p1| + p_3$ as long as $\beta \geq \max\{\beta^\dagger, \beta^{\dagger\dagger}\}$, which implies $A_{\beta,i}$ will stay negative but increasing when $\beta$ is large enough.

Combining all the discussion above, as long as $\beta \geq \bar{\beta} := \{\beta', \beta'', \beta^\dagger, \beta^{\dagger,\dagger}\}$, we have:

- $\forall i \in \mathcal{I}^*$, $A'_{\beta,i} < 0$, therefore, $A_{\beta,i} > 0$ but decreases as $\beta$ increases;

- $\forall i \notin \mathcal{I}^*$, $A'_{\beta,i} > 0$, therefore, $A_{\beta,i} < 0$ but increases as $\beta$ increases;

Combining with Eq. (14), we have $\tilde{\pi}_{\mathcal{I}^*}^\beta - \pi_{\mathcal{I}^*}$ is decreasing as $\beta$ increases when $\beta \geq \bar{\beta}$. $\qquad\square$

# D  DETAILS ABOUT OTHER PASS@K OPTIMIZATION

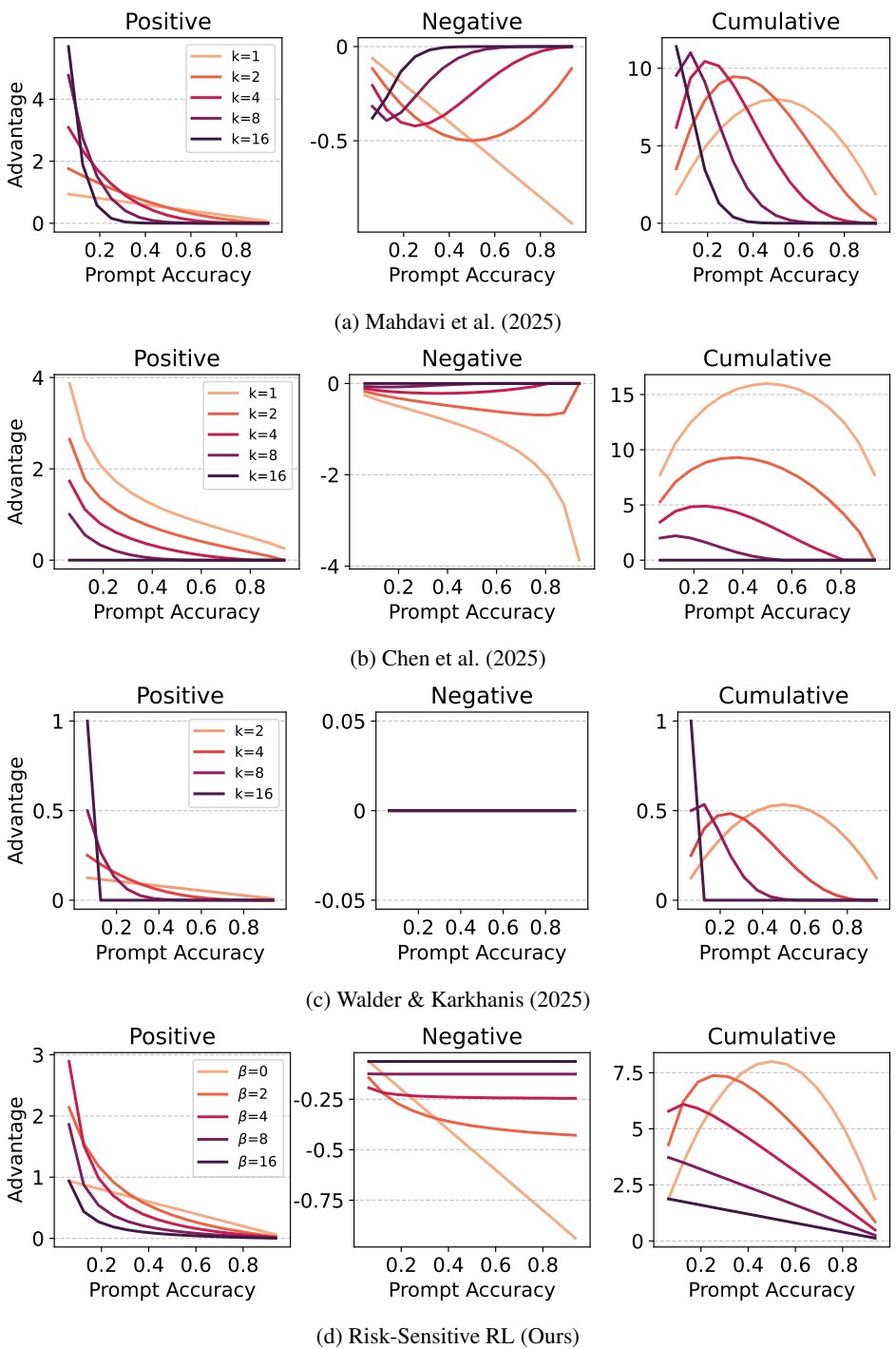

Figure 9: Comparison of advantage estimations across different inference-time objective methods under the binary reward setting with $N = 16$. **Left - Positive**: Advantage estimation for positive responses. **Middle - Negative**: Advantage estimation for negative responses. **Right - Cumulative**: Cumulative absolute advantage value per prompt.

We compare different methods in a binary reward setting (i.e., $r \in \{0, 1\}$). For a given prompt that generates $N$ responses, let $\bar{r} \in [0, 1]$ be the mean reward and $\sigma(r)$ be its standard deviation.

We define the positive advantage, $\hat{A}_{pos}$, as the advantage for responses with a reward of 1, and the negative advantage, $\hat{A}_{neg}$, for those with a reward of 0. The cumulative advantage is the sum of the absolute advantage values over all $N$ responses.

Mahdavi et al. (2025) employs a reweighting policy gradient in the context of pass@k optimization, under the assumption of binary rewards (0 or 1). The expressions for the positive and negative advantages are as follows:

$$\hat{A}_{pos} = k(1 - \bar{r})^k$$
$$\hat{A}_{neg} = -k(1 - \bar{r})^{k-1}\bar{r}$$

(17)

However, as $k$ increases and the prompt accuracy $\bar{r}$ approaches 1, $(1 - \bar{r})^k$ rapidly tends to 0, causing the positive advantage to vanish, as illustrated in the Cumulative column of Fig. 9.

Chen et al. (2025) proposes a pass@k training objective, adopting the binary reward assumption. Among the $N$ responses, let $N_{\text{neg}}$ be the number rewarded with 0 and $N_{\text{pos}} = N - N_{\text{neg}}$ be the number rewarded with 1. Define $\bar{r}_k = 1 - \dfrac{\binom{k}{N_{\text{neg}}}}{\binom{k}{N}}$. The objective is defined by the following positive and negative advantage values:

$$\hat{A}_{pos} = (1 - \bar{r}_k)\sigma(\bar{r}_k)^{-1}$$
$$\hat{A}_{neg} = \left(1 - \bar{r}_k - \frac{\binom{N_{\text{neg}}-1}{k-1}}{\binom{N-1}{k-1}}\right)\sigma(\bar{r}_k)^{-1}$$

(18)

Tang et al. (2025) introduces a best-of-N training objective and utilizes a leave-one-out strategy to reduce the variance of the policy gradient. The advantage $\hat{A}_i$ for this objective is defined as:

$$\hat{A}_i = \max_{\substack{j \in \mathcal{I} \\ \mathcal{I}=\{1,2,\dots,N\}}} r(x, y_j) - \max_{\substack{j \in \mathcal{I} \\ \mathcal{I}=\{1,2,\dots,N\}\setminus\{i\}}} r(x, y_j)$$

(19)

Walder & Karkhanis (2025) builds upon the work of Tang et al. (2025) and further generalizes the method to a smoothed maximum objective, where $k < N$. This generalization involves considering the maximum reward within subsets of size $k$. The policy gradient for this smoothed objective is given by:

$$\hat{A}_i = \frac{1}{\binom{N-1}{k-1}} \sum_{\substack{|\mathcal{I}|=k \\ i \in \mathcal{I} \\ \mathcal{I} \subseteq \{1,2,\dots,N\}}} \left( \max_{j \in \mathcal{I}} r(x, y_j) - \max_{j \in \mathcal{I}\setminus\{i\}} r(x, y_j) \right).$$

(20)

Walder & Karkhanis (2025) and Chen et al. (2025) investigate pass@k-style objectives estimated by uniformly sampling size-$k$ subsets from the $N$ generated responses and computing the probability that a subset contains at least one correct response. The corresponding advantage is then derived from these subset-based pass@k estimates. For a given prompt, let $\bar{r} \in [0, 1]$ denote the empirical fraction of correct responses among the $N$ candidates; then $N(1 - \bar{r})$ is the number of incorrect responses. Selecting $k > N(1 - \bar{r})$ guarantees that every size-$k$ subset contains at least one correct response. Consequently, the best-of-$k$ score equals 1 for every subset; hence, by Eq. 18 and Eq. 20, the advantage collapses to zero and the gradient vanishes. Therefore, the advantage estimates of these methods are not dense; when $\bar{r} > 1 - \frac{k}{N}$, i.e., the prompt accuracy exceeds $(1 - \frac{k}{N})$, the advantage estimate also becomes zero, as confirmed in Fig. 9.

As illustrated in Fig. 9, we compare various methods based on their positive, negative, and cumulative advantages (the sum of absolute advantage values) in a binary reward setting with $N = 16$. Our approach overcomes two key limitations of prior work. First, methods such as those in (Mahdavi et al., 2025; Chen et al., 2025) are confined to binary rewards and do not naturally extend to continuous reward spaces. Second, in existing pass@k optimization techniques, the advantage estimate vanishes when the sample accuracy exceeds $(1 - \frac{K}{N})$. This limitation is highlighted in the "Cumulative" column of Fig. 9, where the magnitude of the advantage estimate, which dictates the optimization weight, drops to zero.

In our comparative analysis of different methods, we select hyperparameters to ensure a fair comparison. For Risk-Sensitive RL, we set $\beta = 2$, which strikes a balance between pass@1 and pass@k performance. For the baseline methods, we use $k = 4$. This choice is motivated by the observation in the "Cumulative" column of Fig. 9, where the peak advantage for RS-GRPO with $\beta = 2$ is approximately 0.2, which aligns with the peak advantage of other methods when $k = 4$. Our experimental results, as shown in Fig. 6, indicate that the method from Walder & Karkhanis (2025) yields unsatisfactory outcomes. Its advantage estimates are persistently positive, and we observe that this absence of negative advantage leads to rapid entropy collapse and poor training performance, a finding consistent with prior work on the importance of negative advantage (Zhu et al., 2025). While other methods (Chen et al., 2025; Mahdavi et al., 2025) achieve pass@32 performance comparable to RS-GRPO, their pass@1 performance is substantially lower. This highlights the benefit of the denser advantage signals provided by RS-GRPO.

# E  ADDITIONAL EXPERIMENTS

## E.1  ENTROPY ANALYSIS

We investigate the connection between entropy changes and Risk-sensitive RL. As shown in Section 4.3, a larger $\beta$ value typically leads to a higher cumulative solve rate on the training set and encourages stronger exploration. Figure 4.3 illustrates the entropy loss dynamics during training. Our findings indicate that while a correlation exists, entropy does not consistently increase with larger $\beta$ values. This suggests that entropy loss may be a biased indicator and might not fully capture the extent of exploration. Moreover, we observe a relationship between optimizing the risk-seeking objective and an increase in entropy, as evidenced by the lowest entropy levels occurring when $\beta = 0$.

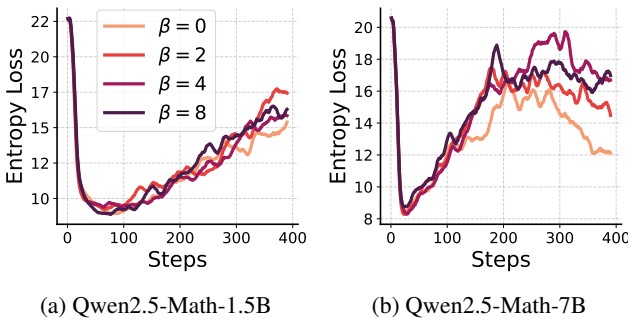

(a) Qwen2.5-Math-1.5B      (b) Qwen2.5-Math-7B

Figure 10: Entropy Analysis under RS-GRPO Training with Different $\beta$ Values

## E.2  ABLATION OF RESPONSES PER PROMPT N

Unless otherwise stated, all reported experiments use rollout $N = 16$, i.e., responses per prompt $N = 16$. In this section, we ablate the effect of responses per prompt $N$ on model performance. We fine-tune Qwen2.5-7B-MATH on the dapo17k dataset and plot training curves of the average pass@$k$ for $k \in \{1, 2, \ldots, 32\}$ across five benchmarks (AIME24, AIME25, CMIMC25, HMMT Feb-24, HMMT Feb-25). Because MATH500 contains many problems and computing pass@32 is costly, we exclude MATH500 from these training checkpoints. Hyperparameter settings follow Table 5. We keep the total batch size fixed at $512 \times 16$; thus for $N = 8$ the training batch size is 1024, for $N = 4$ it is 2048, and for $N = 32$ the prompt batch size is 256. We also adopt dynamic sampling so that each prompt has neither all-correct nor all-incorrect accuracy, ensuring non-zero gradients. Our results show that RS-GRPO ($\beta = 2$) consistently outperforms GRPO ($\beta = 0$) across all choices of $N$, delivering higher pass@2 through pass@32, which further demonstrates the robustness of our method.

## E.3  COMPREHENSIVE PASS@K COMPARISON WITH BASELINES

We train Qwen2.5-Math-7B on the deepmath103k dataset and plot the evolution of the average Pass@$\{1, ..., 32\}$ across five benchmarks (AIME24, AIME25, CMIMC25, HMMT_Feb24,

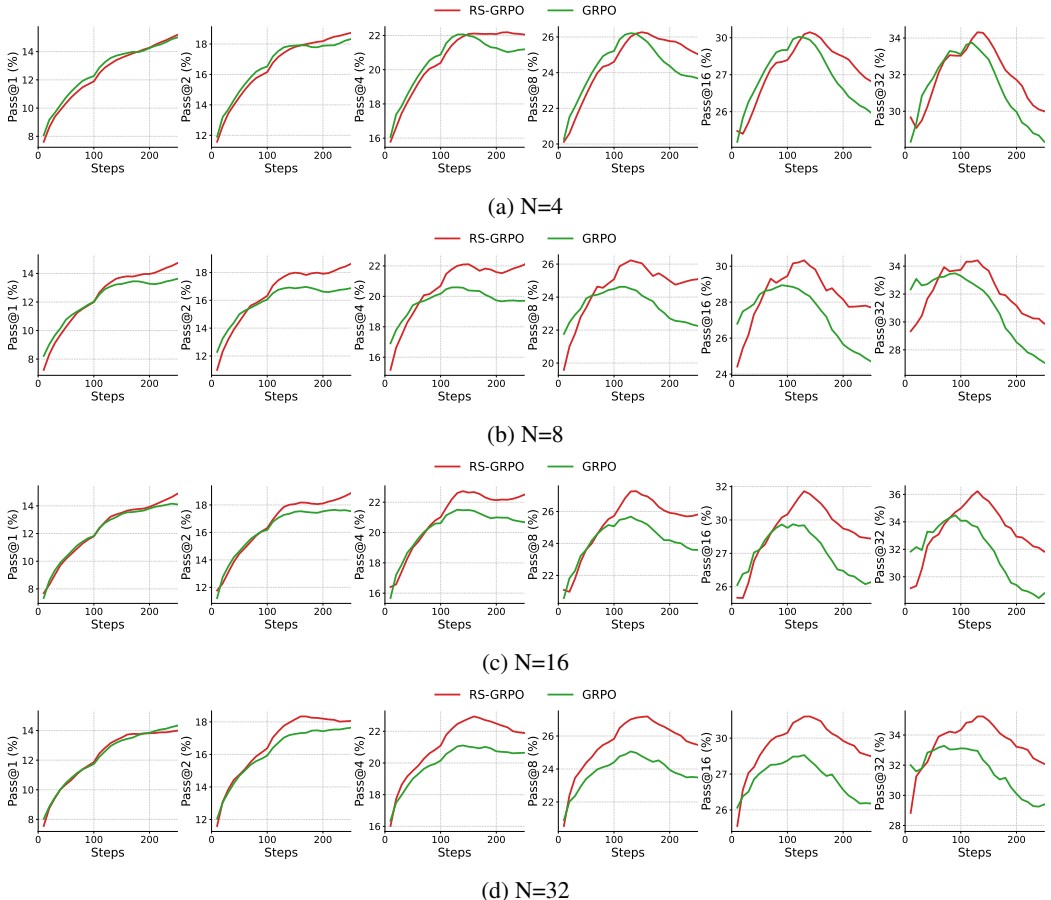

Figure 11: Ablation the effect of responses per prompt (N). We vary $N \in \{4, 8, 16, 32\}$ while keeping the total batch size fixed, and report average pass@$k$ for $k \in \{1, 2, \ldots, 32\}$ across AIME24/25, CMIMC25, and HMMT Feb-24/Feb-25 using Qwen2.5-7B-MATH trained on dapo17k. RS-GRPO ($\beta = 2$) consistently outperforms GRPO ($\beta = 0$) for all $N$, highlighting the robustness of the method.

HMMT_Feb25) during training. RS-GRPO uses $\beta = 2$, and Pass@$k$ baselines (Chen et al., 2025; Mahdavi et al., 2025) use k=4, which is a fair comparison discussed in Appendix D. Compared to Pass@k methods, only RS-GRPO surpasses GRPO on Pass@1, and RS-GRPO shows stronger performance for Pass@$\{1...8\}$ against other baselines, highlighting the benefit of its denser advantage, which better balances Pass@1 and Pass@k.

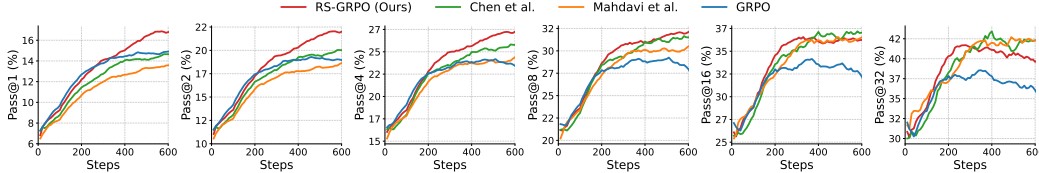

Figure 12: Comprehensive Pass@k comparison with baselines.

# F    IMPLEMENTATION DETAILS

**Datasets**    We trained our models using the following datasets from Hugging Face:

- math12k (Cobbe et al., 2021): `hiyouga/math12k`

- dapo17k (Yu et al., 2025): `BytedTsinghua-SIA/DAPO-Math-17k`
- deepmath103k (He et al., 2025b): `zwhe99/DeepMath-103K`

For evaluation, we used the following datasets, also from Hugging Face:

- MATH500: `math-ai/math500`
- AIME24: `HuggingFaceH4/aime_2024`
- AIME25: `math-ai/aime25`
- HMMT_Feb24: `MathArena/hmmt_feb_2024`
- HMMT_Feb25: `MathArena/hmmt_feb_2025`
- CMIMC25: `MathArena/cmimc_2025`

**Training Details**   Our implementation is based on the VeRL framework (Sheng et al., 2024), and we utilize vLLM 0.8.5 (Kwon et al., 2023) for our experiments. During reinforcement learning training, we do not apply KL regularization. The maximum response length (in tokens) varies by model: 3,072 for Qwen2.5-Math-1.5B and Qwen2.5-Math-7B, and 8,192 for Qwen2.5-7B, Qwen3-4B, and Llama-3.1-8B-Instruct. We use Math_Verify[2] as the ground-truth reward model (reward = 1 for a correct answer, 0 otherwise). For every question, we append the string `\nPlease reason step by step, and put your final answer within \boxed{}` as the prompt.

Table 5 summarizes the hyperparameters employed in our experiments. All the experiments keep these identical. For the experiments in Fig. 4, we set $\beta = 8$ for RS-GRPO. For the comparison with other pass@k methods in Tab. 2, we set $k = 4$ for all pass@k methods and $\beta = 2$ for RS-GRPO. This comparison is fair, as further discussed in Sec. D.

Table 5: **Hyperparameters used in our experiments During RL Training.**

| Hyperparameter | Value |
| --- | --- |
| Temperature | 1.0 |
| Top-p | 1.0 |
| learning rate | $1 \times 10^{-6}$ |
| Training prompt batch size | 512 |
| Responses per prompt $N$ | 16 |
| PPO epochs | 1 |
| PPO mini-batch size | 32 |
| PPO clip_high | 0.28 |
| PPO clip_low | 0.2 |
| Entropy loss coefficient | 0 |
| KL coefficient | 0 |

**Evaluation Details**   The MATH500 benchmark contains 500 problems, while the other datasets consist of 30 or 40 problems each. During inference, we set the sampling temperature to 1.0 and use a top-p value of 0.7. For most benchmarks, we generate $N = 1024$ candidate solutions per problem. However, for the larger MATH500 dataset, we use $N = 32$ to ensure the evaluation remains computationally feasible. For the training curve metrics recording (like Figure 6 and 5), we set $N = 1$ for MATH500 and $N = 32$ for the other datasets. Thus, the testing pass@1 metric records the average across 6 benchmarks, and the testing pass@32 metric records the average across the 5 benchmarks excluding MATH500. We compute the pass@k metric using the unbiased estimator proposed in (Chen et al., 2021).

```
def pass_at_k(n, c, k):
    """
    :param n: total number of samples
    :param c: number of correct samples
    :param k: k in pass@k
```

---

[2]`https://github.com/huggingface/Math-Verify`

```
"""
if n - c < k: return 1.0
return 1.0 - np.prod(1.0 - k /
    np.arange(n - c + 1, n + 1))
```

**Bandit Experiment Details**  In the experiments of Section 3, we consider a bandit setting with 100 actions, denoted as $\mathcal{A} = \{a_1, \ldots, a_{100}\}$. We employ a softmax policy $\pi_\theta$ parameterized by $\theta \in \mathbb{R}^{100}$, where $\theta = [\theta_1, \ldots, \theta_{100}]$. The probability of selecting action $a_i$ is given by:

$$\pi_\theta(a_i) = \frac{e^{\theta_i}}{\sum_{j=1}^{100} e^{\theta_j}}, \quad \text{for all } i \in [100].$$

For each stochastic policy gradient update, we set the batch size $N = 16$ and the learning rate to 0.1.

