# OpenReview forum: "Risk-Sensitive Reinforcement Learning for Alleviating Exploration Dilemmas in Large Language Models"
_ICLR.cc/2026/Conference — ICLR 2026 Poster_

### Official Review · Reviewer_586J · 2025-10-28

**Soundness:** 3
**Presentation:** 3
**Contribution:** 2
**Rating:** 6
**Confidence:** 2

**Summary:**

This paper proposes a risk‑sensitive RL objective for LLM fine-tuning with verifiable rewards. Concretely, the authors replace the mean-reward objective with the exponential-utility formulation that interpolates between mean and max reward, yielding a simple "drop‑in" advantage estimator for GRPO. The paper argues this encourages exploration from sharply peaked pretrained policies and improves pass@k while preserving pass@1. Empirically, the method is evaluated on six math benchmarks and several LLMs, reporting consistent pass@k gains and mostly non‑degrading pass@1. A small bandit study and lemmas provide intuition/theory.

**Strengths:**

1. Clear, readable paper with helpful figures. The exposition around the exploration dilemma and how the exponential utility shifts gradient mass toward hard prompts is crisp. The overview figure and the derivation culminating in Equaqtion 7 are easy to follow.

2. RS‑GRPO only alters the advantage computation, making it nearly "plug‑and‑play" for GRPO pipelines. This is appealing for practitioners.

3. The study covers multiple LLMs and benchmarks; pass@k curves across k=1..1024 are shown in Fig. 4, and $\beta$‑ablations (Fig. 5) help quantify the trade‑off between exploration and convergence.

4. Lemmas in section3.2 show that, in a bandit, risk-neutral PG can reduce probability on the optimal arm (Lemma 2), whereas risk‑sensitive PG increases it for sufficiently large $\beta$ (Lemma 3), with diminishing gains if $\beta$ is too large (Lemma 4). The toy bandit (Fig. 3) visually supports this.

**Weaknesses:**

1. Limited novelty. The exponential-utility objective and risk‑sensitive policy gradients are classical (e.g., Howard & Matheson; Mihatsch & Neuneier), and the paper essentially instantiates that objective as a reweighted advantage for GRPO (Eq. 7-8). The algorithmic delta is small and conceptually straightforward---primarily a different per‑sample weighting. Theoretical results are restricted to a bandit setting; there is no analysis in sequence‑generation or RLVR settings beyond intuition. This makes the contribution marginal.

   - Table 2 (p.8) shows RS‑GRPO usually adds ~+3–5 points in pass@32 over GRPO (sometimes less), while pass@1 is often flat and occasionally lower. For instance, on Llama‑3.1‑8B‑Instruct AIME24, pass@1 drops from 9.9 (GRPO) to 9.4 (RS‑GRPO); on AIME25 it drops from 1.2 to 0.6. The Qwen2.5‑Math‑1.5B MATH500 pass@1 also decreases slightly (80.2$\rightarrow$78.1). These are small but non‑negligible regressions.

2. The comparison to pass@k optimization baselines uses k=4 for those methods and $\beta$=2 for RS‑GRPO. Although the paper argues this equalizes the advantage magnitude (Fig. 8), the mapping between k and $\beta$ is not unique. Baselines might benefit from a small hyperparameter sweep; it is unclear if they were tuned comparably to RS‑GRPO.

3. There are some design choices that look unnatural to me.

   - No KL regularization is used during RL (Table 3). This can lead to policy drift; evaluating safety, verbosity, or formatting regressions would strengthen the case that risk‑seeking exploration is benign beyond accuracy.

   - Risk‑seeking emphasizes tail rewards and may amplify false positives from the verifier. There is no experiment where the reward is intentionally perturbed or partially noisy to test robustness.

   - Many benchmarks use N=1024 samples per problem at evaluation. It would help to quantify gains at small k (e.g., k in {4,16}) under fixed compute budgets and to report training FLOPs/tokens; the training curves (Fig. 5-6) stop at a few hundred steps but do not translate to compute cost.

4. There is a gap between the theory statements and experiments. The claims are theoretically justified only for bandits and do not consider sequence-level credit assignment, KL penalties, or importance sampling present in practical RLVR. Extending the analysis (even approximately) to token-level updates or to a simple MDP with stochastic rewards would increase significance.

**Questions:**

1. Did you sweep k for the pass@k baselines and report the best? Why is k=4 the sole setting (Appendix D)? Could stronger tuning reverse the small pass@1 gap you observe in Fig. 6?

2. What are the training FLOPs / tokens and wall‑clock for RS‑GRPO vs GRPO? How do gains scale with fewer rollouts (smaller N) or shorter training?

3. You set KL coefficient to 0 (Table 3). Do results hold with a modest KL (as in many RLHF pipelines)? How sensitive are the conclusions?

---

> ### Author Response · Authors · 2025-11-21
> **Authors' Rebuttal Part 1**
>
> We thank the reviewer for the valuable feedback, and we address your concerns below.
>
> ### W1: Limited novelty.
>
> Regarding novelty, we agree with the reviewer that “the exponential-utility objective and risk‑sensitive policy gradients are classical”, **but we are the first to incorportate this idea into LLM post-training and demonstrate its effectiveness**, which we believe are novel contributions to this community. Besides, **prior work[1,2] has observed that LLM reinforcement learning tends to sharpen the policy, and we provide a theoretical (Section 3.2)  and empirical  explanation (Section 3.1) for this phenomenon:** when starting from a sharply peaked initial LLM policy, optimizing only the mean reward naturally leads to further policy sharpening while abandoning exploration of potentially better solutions. In contrast, Risk-Sensitive RL provably increases the probability mass on optimal actions.
>
> **Although our analysis is conducted in the bandit setting, we would like to remind the reviewer that the widely-used GRPO algorithm is fundamentally modeled as a bandit problem** (since only the end of the response receives a reward, the advantage is uniform across the entire response). Therefore, there is virtually no gap between our theoretical analysis and practical implementation.
>
> While some individual metrics show slight decreases, **the overall results in Table 2 demonstrate comprehensive improvements: 0–3% gains in pass@1 and 2–5% gains in pass@32.** Notably, integrating RS-GRPO with GRPO requires modifying only a few lines of code with no additional computational overhead—we provide the implementation in our supplementary materials. **Achieving these improvements with such minimal modifications is significant.**
>
> ### W2 & Q1: Pass@k baseline performance may be underestimated.
>
> We remind the reviewer that we follow [3,4] in setting our baseline hyperparameter k. **$k=4$ is the default and recommended setting in papers [3,4]**. As the reviewer noted, our method is primarily a different per-sample weighting scheme. We believe that comparing methods at the arguments of the maxima of per-sample weighting is fair ($\beta=2$ vs $k=4$). Given the significant computational cost of running LLM experiments, we hope the reviewer understands that we cannot sweep the parameter k.
>
> ### W3(part1) & Q3: No KL regularization is used during RL.
>
> Since our focus is on reinforcement learning from verifiable rewards (RLVR), we drop KL regularization by default, which is also the mainstream configuration in RLVR recently. We appreciate this suggestion and have added an experiment comparing training with and without KL regularization in Section 4.8 (marked in blue). The results are shown in Figure 8.
>
> Figure 8 shows that RS-GRPO+KL consistently outperforms GRPO+KL across Pass@{1,2,…,32}, though the KL term slightly reduces the overall performance ceiling.
>
> ### W3(part2):  There is no experiment where the reward is intentionally perturbed or partially noisy to test robustness.
>
> We thank the reviewer for raising this important question. However, our focus is on reinforcement learning from verifiable rewards (RLVR), which applies to most reasoning tasks such as mathematics and coding. In these settings, the reward signal is reliable. While noisy rewards may indeed affect the performance of risk-seeking methods, we believe investigating this scenario is beyond the scope of this paper.
>
> ### W3(part2) & Q2:  training FLOPs for RS‑GRPO vs GRPO;  Gains scale with smaller N and smaller k (eg, 4,8,16)
>
> As we highlighted in the “Practical Implementation” part in Section 2.1, RS-GRPO only alters the advantage computation (i.e. the reward transformation), and the additional computational overhead can be ignored and thus training FLOPs  are almost same.
>
> All RS-GRPO and GRPO performance metrics are compared at the same number of training steps. For a longer training comparison of Pass@{1…32}, see Figure 12 (Appendix E.3). RS-GRPO shows stronger performance for Pass@{1...8} against other baselines.

---

> ### Author Response · Authors · 2025-11-21
> **Authors' Rebuttal Part2**
>
> ### W4: There is a gap between the theory statements and experiments
>
> We believe extending our theory to multi-turn setup is a valuable direction, but it is orthogonal to the scope of this paper and we leave it for future work.
>
> Although our analysis is conducted in the bandit setting, we would like to remind the reviewer that the widely-used GRPO algorithm is fundamentally modeled as a bandit problem (since only the end of the response receives a reward, the advantage is uniform across the entire response). Therefore, there is virtually no gap between our theoretical analysis and practical implementation.
>
> Empirically, we note that in many existing RLVR agentic (multi-turn) setups (e.g. coding agent and search agent), the only reward signal is the verifiable reward at the terminal step. In those cases, our method applies directly, and we expect it yields performance gains. We have added this remark in Appendix A (Limitation).
>
> ### Reference
>
> [1] Does reinforcement learning really incentivize reasoning capacity in llms beyond the base model?  NeurIPS 2025.
>
> [2] The Invisible Leash: Why RLVR May Not Escape Its Origin.
>
> [3] Pass@k training for adaptively balancing exploration and exploitation of large reasoning models.
>
> [4] Pass@K Policy Optimization: Solving Harder Reinforcement Learning Problems.
>
> ---
>
> We would like to thank you again for raising important questions about this work, and we believe the additional clarifications have strengthened the paper. Please let us know if you have any additional concerns or questions.

---

> > ### Comment · Reviewer_586J · 2025-11-22
> >
> > Thanks for the detailed response. My questions (W2, W3, W4) have been addressed. I will maintain my positive score.

---

### Official Review · Reviewer_aF1e · 2025-10-30

**Soundness:** 3
**Presentation:** 3
**Contribution:** 3
**Rating:** 6
**Confidence:** 4

**Summary:**

The paper directly addresses a critical issue in Reinforcement Learning with Verifiable Rewards (RLVR) for LLMs. The exploration dilemma, where LLMs get stuck in suboptimal solutions due to their pre-trained biases, is clearly articulated.

**Strengths:**

1. Introduction of a risk-sensitive RL framework for LLM fine-tuning.
2.Theoretical and empirical support for the risk-sensitive approach.
3. Demonstration of significant improvements in pass@k performance with maintained pass@1 accuracy.

**Weaknesses:**

1.The paper focuses on mathematical reasoning. Investigating the applicability of RS-GRPO to other OOD tasks is necessary.
2. While the paper presents an entropy analysis, it acknowledges that entropy loss may be a biased indicator of exploration. Exploring alternative metrics for quantifying exploration in this context could be beneficial.
3. The paper notes that for some models (e.g., Qwen2.5-7B, Llama3.1-8B-Instruct), RS-GRPO failed to outperform the base model at high values of *k*. Further investigation into the reasons for this behavior could lead to more robust performance across different LLM architectures.
4. A key limitation is the use of a fixed risk-seeking parameter.

**Questions:**

/

---

> ### Author Response · Authors · 2025-11-21
>
> We thank the reviewer for the valuable feedback, and we address your concerns below.
>
> ### W1: Investigating the applicability of RS-GRPO to other OOD tasks is necessary.
>
> We directly evaluated Qwen2.5-Math7B models trained on the deepmath103 dataset (whose math performance was already reported in Table 2) on GPQA-Diamond (Rein et al., 2024), which covers biology, physics, and chemistry. The results show that RS-GRPO consistently outperforms GRPO from pass@1 to pass@32, aligning with our findings on math tasks.
>
> | **模型** | **pass@1** | **pass@2** | **pass@4** | **pass@8** | **pass@16** | **pass@32** |
> | --- | --- | --- | --- | --- | --- | --- |
> | **Qwen2.5-MATH-7B**  | 4.36 | 8.36 | 15.5 | 27.3 | 44.5 | 64.6 |
> | **GRPO** | 41.2 | 52.2 | 62.8 | 71.6 | 78.5 | 84.8 |
> | **RS-GRPO** | 41.8 | 54.1 | 65.5 | 75.2 | 83.1 | 88.9 |
>
> ### W2:  Exploring alternative metrics for quantifying exploration in this context could be beneficial
>
> We remind the reviewer that we already use **cumulative solve rate as a metric for quantifying exploration (Figure 5)**. Cumulative solve rate means that if a problem has at least one correct response, we consider it solved. It represents the proportion of solved problems in the training set. Figure 5 shows that as $\beta$ increases in RS-GRPO, the cumulative solve rate also increases, which aligns with our theoretical expectations.
>
> We acknowledge that the definition of cumulative solve rate was somewhat unclear. We have improved several descriptions in the paper to make this clearer.
>
> ### W3: The paper notes that for some models (e.g., Qwen2.5-7B, Llama3.1-8B-Instruct), RS-GRPO failed to outperform the base model at high values of *k*. Further investigation into the reasons for this behavior could lead to more robust performance across different LLM architectures.
>
> We attribute this to the fact that we evaluate on the mathmatical benchmark. For Qwen2.5-7B and Llama3.1-8B-Instruct, the initial policy distribution may be considerably distant from the optimal policy distribution required by mathmatical benchmark. Consequently, RS-GRPO fails to outperform the base model at high values of *k*—in other words, RS-GRPO struggles to escape the gravitational pull of the initial policy distribution when the optimal policy is far away. In contrast, Qwen3-4B-Base and Qwen2.5-7B-MATH have clearly undergone more extensive performance tuning on math domain, so their initial policy distributions are closer to the optimal policy. As a result, RS-GRPO is able to surpass the base model at high values of *k*. We also note that regardless of the base model choice, RS-GRPO consistently outperforms GRPO at high values of k.
>
> ### W4: A key limitation is the use of a fixed risk-seeking parameter.
>
> As noted in Appendix A (Limitations), we experimented extensively with dynamically adjusting the risk-seeking parameter, but the Pass@1 results consistently underperformed compared to using a fixed, well-chosen beta within the same training budget. *Based on our current findings, we suggest that a fixed risk-seeking parameter (i.e., $\beta=2$) may provide relatively better performance, though further investigation could be valuable.*
>
> Reference:
>
> [1] Pass@K Policy Optimization: Solving Harder Reinforcement Learning Problems. NeurIPS 2025.
>
> [2] Pass@ k training for adaptively balancing exploration and exploitation of large reasoning models.
>
> ---
>
> We would like to thank you again for raising important questions about this work, and we believe the additional clarifications have strengthened the paper. Please let us know if you have any additional concerns or questions.

---

### Official Review · Reviewer_mSfu · 2025-10-30

**Soundness:** 3
**Presentation:** 3
**Contribution:** 2
**Rating:** 6
**Confidence:** 4

**Summary:**

This paper introduces RS-GRPO, a method to address the "exploration dilemma" in RL fine-tuning of LLMs. It argues that standard RL improves pass@1 at the cost of pass@k by overly sharpening the policy. RS-GRPO uses a risk-seeking objective to encourage exploration of diverse solutions, aiming to improve pass@k while maintaining strong pass@1 performance.

**Strengths:**

1. Applying the established theory of Risk-Sensitive RL to LLM fine-tuning is a well-motivated.

2. Most claims are supported with both theoretical analysis and comprehensive experiments across multiple models and benchmarks.

**Weaknesses:**

Despite its strengths, the paper has several weaknesses that temper its overall contribution:
1. Unclear Improvement and Potentially Unfair Comparisons: The central claim of improvement is not consistently strong when viewed in context.
- If the main goal is to improve pass@k, the experimental results (Table 2, Figure 6) show that RS-GRPO performs comparably, and sometimes worse than, other baseline methods designed specifically for pass@k optimization.
- More importantly, the comparison to other pass@k methods may be unfair. Other methods like [1, 2] often employ a two-stage training strategy: first, they run pass@k training to enhance exploration and diversity, and then they perform a second stage of fine-tuning to distill this diversity into a strong pass@1 policy. This paper compares its single-stage method against only the first stage of these baselines, which may not represent their full potential, especially on the pass@1 metric.
2. Unverified Advantage on Continuous Rewards: The paper claims the ability to handle continuous rewards as an advantage, but all experiments use binary rewards, leaving this claim empirically unsubstantiated.
3. New Hyperparameter: The method introduces a hyperparameter, β, which requires tuning and adds complexity for practical application.

**Questions:**

1. Regarding the comparison with other pass@k methods: Could you comment on the fairness of comparing your single-stage method against baselines [1, 2] that often benefit from a second, pass@1-focused fine-tuning stage? Would applying a similar two-stage approach with RS-GRPO provide a more direct and fair comparison of the methods' full capabilities?
2. Regarding the continuous rewards: The ability to handle continuous rewards is presented as a significant advantage. Could you provide any experimental results, even on a smaller scale or a synthetic task, to demonstrate the practical benefits of RS-GRPO in such a setting?
3. The paper's motivation relies on the premise that pre-trained LLMs have a "sharply peaked" and "suboptimal" initial policy. Could you provide more direct evidence to support this? For instance, an analysis of the policy entropy or showing that the most probable reasoning paths are indeed suboptimal would greatly strengthen the paper's foundation.

[1] Chen, Z., Qin, X., Wu, Y., Ling, Y., Ye, Q., Zhao, W. X., & Shi, G. (2025). Pass@ k training for adaptively balancing exploration and exploitation of large reasoning models. arXiv preprint arXiv:2508.10751.

[2] Li, Q., Xue, R., Wang, J., Zhou, M., Li, Z., Ji, X., ... & Yang, J. (2025). Cure: Critical-token-guided re-concatenation for entropy-collapse prevention. arXiv preprint arXiv:2508.11016.

---

> ### Author Response · Authors · 2025-11-21
>
> We thank the reviewer for the valuable feedback, and we address your concerns below.
>
> ### W1 & Q1: Unclear Improvement and Potentially Unfair Comparisons.
>
> Regarding the main goal, our objective is to improve pass@1 performance while ensuring that pass@k does not degrade compared to the Base Model. Previous pass@k optimization methods [1,2,3] can enhance pass@k performance but **often sacrifice pass@1 performance.** We find that our method, due to its denser advantage signals, not only improves pass@k effectiveness but also provides gains in pass@1 compared to GRPO.
>
> In Figure 12 (Appendix E.3), we demonstrate that RS-GRPO ($\beta$=2) outperforms the pass@k (k=4) method, showing performance gains across pass@{1,2,4,8}, which validates the superiority of our approach.
>
> Although one could prevent pass@1 degradation by training with a pass@k objective in stage 1 and then training with pass@1 in stage 2 [1], **even with this two-stage approach, pass@1 performance still does not surpass that of continuously training for pass@1** [1]. While [3] demonstrated the effectiveness of pass@k + pass@1 training on puzzle tasks, it has not been validated on mainstream math and coding benchmarks. As mentioned in our Limitation section (Appendix A), we extensively experimented with various methods for dynamically adjusting $\beta$, but without exception, the final pass@1 performance could not match that achieved with a fixed $\beta$=2 under the same training budget.
>
> [4] involves changes to the rollout strategy, which falls outside our comparison scope. Our focus is on comparing objective function modifications that require only a few lines of code change.
>
> [1] Pass@K Policy Optimization: Solving Harder Reinforcement Learning Problems. NeurIPS 2025.
>
> [2] Beyond Accuracy: A Policy Gradient Reweighting Approach for Pass@K Maximization in LLMs.
>
> [3] Pass@ k training for adaptively balancing exploration and exploitation of large reasoning models.
>
> [4] Cure: Critical-token-guided re-concatenation for entropy-collapse prevention.
>
> ### W2 & Q2: Regarding the continuous rewards
>
> We kindly refer the reviewer to our bandit experiments in Section 3.1, where we consider the continuous rewards.
>
> ### W3: The method introduces a hyperparameter, β, which requires tuning and adds complexity for practical application.
>
> We conducted a beta ablation study in Section 4.3 and found that $\beta=2$ offers an effective trade-off, achieving strong pass@k performance while simultaneously enhancing pass@1. All RS-GRPO results in Table 2 are presented with $\beta=2$.
>
> Additionally, we added experiments on GPQA-Diamond and LiveCodeBench v5, both demonstrating that $\beta=2$ generalizes well across different tasks. And our method requires modifying only a few lines of code on top of GRPO, making it extremely simple to implement.
>
> ### Q3: Provide more direct evidence to support “Pre-trained LLMs have a "sharply peaked" and "suboptimal" initial policy”
>
> - **Sharply peaked distribution.** We use the Qwen2.5-7B model as an example. The vocabulary contains over 150,000 tokens. We randomly sampled 1,000 generated responses and computed the average top-3 token probability distribution during generation. As shown in the table below, the top-1 token (out of 150,000) accounts for approximately 90% of the probability mass, demonstrating a sharply peaked distribution.
>
>
>     | top-1 token prob | top-2 token prob | top-3 token prob |
>     | --- | --- | --- |
>     | 0.935 | 0.042 | 0.011 |
> - **Suboptimal.** In Table 2, we report the Base Model's pass@1 performance. The results show substantial room for improvement, demonstrating that the LLM policy is suboptimal.
>
> ---
>
> We would like to thank you again for raising important questions about this work, and we believe the additional clarifications have strengthened the paper. Please let us know if you have any additional concerns or questions.

---

> > ### Comment · Reviewer_mSfu · 2025-11-22
> >
> > Thank you for the response. I will maintain my positive score.

---

### Official Review · Reviewer_roMf · 2025-10-31

**Soundness:** 2
**Presentation:** 3
**Contribution:** 3
**Rating:** 6
**Confidence:** 4

**Summary:**

This paper addresses the mismatch between the risk-neutral training objectives commonly used in RLVR (e.g., GRPO) and the risk-sensitive evaluation metrics such as pass@k. The authors propose a risk-seeking RL formulation based on an exponential utility function, leading to a risk-sensitive variant of GRPO, termed RS-GRPO. The paper derives the policy gradient for this formulation and demonstrates its benefits over standard GRPO in both bandit experiments and math reasoning tasks across seven base LLMs, with evaluation over $k \in [1, 1024]$. An ablation study finds that $k=2$ yields a good trade-off across different $k$ values in pass@k.

**Strengths:**

- The paper clearly articulates the problem of mismatch between training and evaluation in RL for LLMs and motivates the use of risk-sensitive objectives.
- The derivation from exponential utility to the risk-sensitive policy gradient is mathematically sound and well presented.
- Section 3 provides an excellent illustrative toy example that clarifies the intuition behind the approach and shows how risk-seeking objectives can encourage better exploration.
- The method addresses the vanishing advantage issue for prior pass@k optimization methods.
- Extensive experiments across multiple base models and math reasoning benchmarks convincingly demonstrate consistent improvements over GRPO baseline.
- The paper is well written, logically structured, and easy to follow, which is commendable given the technical depth of the topic.

**Weaknesses:**

**Lack of formal connection between risk-seeking objective and pass@k:**
   The paper repeatedly motivates its approach by the mismatch between risk-neutral objectives and pass@k evaluation, yet it provides no theoretical derivation linking the risk-seeking objective to pass@k performance. While the intuition that risk-seeking encourages exploration is sound, a formal connection would substantially strengthen the argument.

**Lack of formal understanding for the vanishing-advantage analysis:**
   Figure 8 shows the “vanishing advantage”. A formal proof or at least a more detailed analytical explanation of why prior works have this issue.

**Fairness of baseline comparison:**
   The paper compares to baselines trained with $k=4$ but evaluates at $k=1$ and $k=32$. Since prior works *directly* optimize pass@k, such comparison can be misleading. The authors should either match $k$ values during training or explicitly state that the comparison involves different optimization targets. Additionally, analysis of sensitivity to $N$ (number of samples) is needed, as the claimed robustness of RS-GRPO to vanishing advantage depends on this factor.

**Scope:**
   While the method is domain-agnostic as claimed, experiments focus solely on math reasoning. It would be valuable to discuss whether RS-GRPO naturally extends to multi-turn RL in theory and other reasoning domains like coding in practice.

**Questions:**

- Figure 1 lacks clear axis labels and is difficult to interpret.
- Please clarify how “dense signal” in Table 1 is defined.
- What is the “cumulative solve rate” in the training metric?
- In Figure 3, the y-axis should be labeled “reward” rather than “reward distribution.”?

---

> ### Author Response · Authors · 2025-11-21
> **Authors' Rebuttal Part 1**
>
> We thank the reviewer for the valuable feedback, and we address your concerns below.
>
> ### W1: Lack of formal connection between risk-seeking objective and pass@k:
>
> We believe risk-seeking objective and pass@$k$ have connections. The key observation is that both the pass@$k$ and our risk-seeking objective gradually converge to the same maximum reward $\max_y r(y)$, as $k, \beta \rightarrow +\infty$, respectively. In other words, both of them can be regarded as approximation to the maximum reward, which implies the correlation between optimizing our risk-sensitive objective and the pass@$k$. We demonstrate the advantages of the risk-seeking objective compared to the pass@k objective in Table 1.
>
> ### W2: Lack of formal understanding for the vanishing-advantage analysis:
>
> Thank you for this suggestion. We have updated Appendix D (Details About Other Pass@k Optimization), marked in blue, to provide a detailed explanation of why other Pass@k methods suffer from the vanishing-advantage problem. We use [1,2] as examples to illustrate this issue.
>
> [1, 2] investigate pass@k-style objectives estimated by uniformly sampling size-$k$ subsets from the $N$ generated responses and computing the probability that a subset contains at least one correct response. The corresponding advantage is then derived from these subset-based pass@k estimates.
> For a given prompt, let $\bar{r}\in[0,1]$ denote the empirical fraction of correct responses among the $N$ candidates; then $N(1-\bar{r})$ is the number of incorrect responses. Selecting $k > N(1-\bar{r})$ guarantees that every size-$k$ subset contains at least one correct response.
> Consequently, the best-of-$k$ score equals 1 for every subset; hence, the advantage collapses to zero and the gradient vanishes. In other words, when $\bar{r}> 1 - \frac{k}{N}$, i.e., the prompt accuracy exceeds $(1 - \frac{k}{N})$, the advantage estimate also becomes zero, as confirmed in Figure 10.
>
> Reference:
>
> [1] Pass@K Policy Optimization: Solving Harder Reinforcement Learning Problems. NeurIPS 2025.
>
> [2] Pass@ k training for adaptively balancing exploration and exploitation of large reasoning models.
>
> ### W3: Fairness of baseline comparison
>
> - Part1: Compasion of Pass@k={1,2,4,8,16,32} with baselines.
>
> We supplemented Figure 12 in Appendix E.2 with a comparison of **Pass@k={1,2,4,8,16,32} between RS-GRPO and baselines. The results show that** RS-GRPO is the only one that surpasses GRPO on Pass@1, and RS-GRPO demonstrates stronger performance for Pass@{1...8} compared to other baselines, highlighting the benefit of its denser advantage signal, which better balances Pass@1 and Pass@k.
>
> - Part2:  Analysis of sensitivity to rollout N=(4, 8, 16, 32)
>
> **We conducted an ablation study on N ∈ {4, 8, 16, 32} with results presented in Appendix E.2, Figure 11.** The total batch size is fixed at 8192 (512×16) as default. For N=8/4/32, we adjust the prompt batch size accordingly: for N=4, 8, 32, the training batch size are 2048, 1024 and 256, respectively.
>
> **Our key findings show that RS-GRPO ($\beta$=2) consistently outperforms GRPO ($\beta$=0) across Pass@2–Pass@32 for all values of N** throughout the entire training process, demonstrating the robustness of our approach.
>
>
> ### W4: Scope(whether RS-GRPO naturally extends to multi-turn RL in theory and other reasoning domains like coding in practice.)
>
> We believe extending our theory beyond the bandit setup is a valuable direction, but it falls outside the scope of this paper and we leave it for future work.
>
> Empirically, we note that in many existing RLVR agentic (multi-turn) setups (e.g., coding and search agents), the only reward signal is the verifiable reward at the terminal step. In those cases, our method applies directly, and we expect it to yield performance gains. We have added this remark in Appendix A (Limitation).
>
> In terms of coding reasoning tasks, We conducted additional code generation experiments using Qwen2.5-7B-Instruct-1M on the HuggingFace dataset `ganler/code-r1-12k`,  and evaluated on LiveCodeBench v5.  Set max response length to 4k and other hyberparamters keep identical to Table 5. The results show that RS-GRPO($\beta=2$) outperforms GRPO on coding tasks, with a 1.2% improvement in pass@1 and a 4.5% improvement in pass@8. While the Base Model's pass@8 exceeds GRPO's trained result, RS-GRPO's pass@8 still surpasses the Base Model.
>
> |  | **pass@1 (%)** | **pass@2（%）** | **pass@4（%）** | **pass@8（%）** |
> | --- | --- | --- | --- | --- |
> | Base Model | 22.6 | 27.4 | 318 | 35.6 |
> | GRPO | 28.5 | 30.4 | 31.7 | 32.7 |
> | RS-GRPO | 29.7 | 32.5 | 34.6 | 36.2 |

---

> ### Author Response · Authors · 2025-11-21
> **Authors' Rebuttal Part 2**
>
> ### Q1:  Figure 1 lacks clear axis labels and is difficult to interpret.
>
> We have added captions in Figure 1 to make our insight clear.
>
> ### Q2: Please clarify how “dense signal” in Table 1 is defined.
>
> We appreciate this question. We have added further descriptions in Appendix D. The concept of "dense signal" can be understood by examining the Cumulative column in Figure 9 (Appendix). When the prompt accuracy is relatively high, typically when accuracy > $1-\frac{k}{N}$, baseline methods experience a sudden drop to zero advantage for that prompt, whereas our method maintains a non-zero advantage. This is what we refer to as a dense signal. The benefit of maintaining a dense signal is that it enables better balance between pass@1 and pass@k performance. Other methods sacrifice pass@1 performance to improve pass@k, while our approach can achieve gains in both metrics (as shown in Figure 6 and Figure 12).
>
> ### Q3: What is the “cumulative solve rate” in the training metric?
>
> Cumulative solve rate means that if a problem has at least one correct response, we consider it solved. It represents the proportion of solved problems in the training set. We acknowledge that the definition of cumulative solve rate was somewhat unclear. We have improved the descriptions in the paper to make this clear.
>
> ### Q4: In Figure 3, the y-axis should be labeled “reward” rather than “reward distribution.”?
>
> Thank you for this suggestion—we have corrected it.
>
> ---
>
> We would like to thank you again for raising important questions about this work, and we believe the additional clarifications have strengthened the paper. Please let us know if you have any additional concerns or questions.

---

### Meta-Review · Area_Chair_zAmt · 2026-01-07

**Summary:**

This paper proposes a risk-sensitive reinforcement learning framework (RS-GRPO) for LLM post-training with verifiable rewards, aiming to address the exploration dilemma caused by sharply peaked pretrained policies. The reviewers generally agree that the paper is clearly written, technically sound, and well motivated, with a simple and practical algorithmic modification. Extensive experiments across multiple math reasoning benchmarks and base models show consistent improvements in pass@k while mostly maintaining or slightly improving pass@1. The work is seen as particularly appealing from a practical perspective, as RS-GRPO is almost plug-and-play on top of GRPO. The main points of discussion concerned limited novelty, fairness of baseline comparisons, the bandit-only theoretical analysis, and the initially narrow experimental scope.

**Reviewer Concerns:**

Most concerns are addressed by the authors. Novelty still seems to be one concern to Reviewer 586J, although the overall score is positive.

**Reviewer Scores:**

I think all the reviewers will maintain positive attitude to the paper.

---

### Decision · Program_Chairs · 2026-01-26

Accept (Poster)